# Mashes to Mashes, Crust to Crust. Presenting a novel microstructural marker for malting in the archaeological record

**Andreas G. Heiss**[1‡]*, **Marian Berihuete Azorín**[2¤‡], **Ferran Antolín**[3‡], **Lucy Kubiak-Martens**[4‡], **Elena Marinova**[5,6‡]*, **Elke K. Arendt**[7☉], **Costas G. Biliaderis**[8☉], **Hermann Kretschmer**[9☉], **Athina Lazaridou**[8☉], **Hans-Peter Stika**[2‡], **Martin Zarnkow**[10‡], **Masahiro Baba**[11☉], **Niels Bleicher**[12‡], **Krzysztof M. Ciałowicz**[13☉], **Marek Chłodnicki**[14☉], **Irenäus Matuschik**[5☉], **Helmut Schlichtherle**[5☉], **Soultana Maria Valamoti**[15,16‡]

1 Department for Bioarchaeology, Austrian Archaeological Institute (ÖAI), Austrian Academy of Sciences (ÖAW), Wien, Vienna, Austria, 2 Department of Molecular Botany (190a), Institute of Biology, University of Hohenheim, Stuttgart, Germany, 3 Integrative Prehistory and Archaeological Science (IPAS/IPNA), University of Basel, Basel, Switzerland, 4 BIAX Consult, Biological Archaeology & Landscape Reconstruction, Zaandam, The Netherlands, 5 Hemmenhofen Office, State Office for Cultural Heritage Baden-Württemberg, Gaienhofen-Hemmenhofen, Germany, 6 Center for Archaeological Sciences (CAS), KU Leuven, Leuven, Belgium, 7 Cereal and Beverage Science Research Group, School of Food & Nutritional Sciences, University College Cork, Cork, Ireland, 8 Laboratory of Food Chemistry & Biochemistry, Department of Food Science & Technology, Faculty of Agriculture, Aristotle University of Thessaloniki, Thessaloniki, Greece, 9 Braxar GmbH, Bruchsal, Germany, 10 Research Center Weihenstephan for Brewing and Food Quality, Technical University of Munich (TUM), Freising, Germany, 11 Waseda Institute for Advanced Study, Waseda University, Tokyo, Japan, 12 Office for Urbanism Zürich, Underwater Archaeology and Laboratory for Dendrochronology, Zürich, Switzerland, 13 Institute of Archaeology, Jagiellonian University, Kraków/Cracow, Poland, 14 Department for General Archaeology, Poznań Archaeological Museum, Poznań, Poland, 15 Lira Laboratory, Department of Archaeology, School of History and Archaeology, Aristotle University of Thessaloniki, Thessaloniki, Greece, 16 Center for Interdisciplinary Research and Innovation (CIRI-AUTH), Aristotle University of Thessaloniki, Thessaloniki, Greece

☉ These authors contributed equally to this work.
¤ Current address: Catalan Institute of Human Paleoecology and Social Evolution (IPHES), Tarragona, Spain
‡ These authors also contributed equally to this work.
* andreas.heiss@oeai.at (AGH); elena.marinova-wolff@rps.bwl.de (EM)

## Abstract

The detection of direct archaeological remains of alcoholic beverages and their production is still a challenge to archaeological science, as most of the markers known up to now are either not durable or diagnostic enough to be used as secure proof. The current study addresses this question by experimental work reproducing the malting processes and sub-sequent charring of the resulting products under laboratory conditions in order to simulate their preservation (by charring) in archaeological contexts and to explore the preservation of microstructural alterations of the cereal grains. The experimentally germinated and charred grains showed clearly degraded (thinned) aleurone cell walls. The histological alterations of the cereal grains were observed and quantified using reflected light and scanning electron microscopy and supported using morphometric and statistical analyses. In order to verify the experimental observations of histological alterations, amorphous charred objects (ACO) containing cereal remains originating from five archaeological sites dating to the 4th millennium BCE were considered: two sites were archaeologically recognisable brewing

request at the University of Hohenheim, Institute of Botany (2010), Garbenstraße 30, 70593 Stuttgart, Germany. Likewise, all archaeological samples are accessible for scientific re-evaluation on request: The brewing residues from Hierakonpolis are stored at the magazines at the site and can be accessed by obtaining permission from the Egyptian Antiquity Authorities. Likewise, the materials from Tell el-Farkha are accessible for re-evaluation after the aforementioned permissons. The charred cereal products from Hornstaad–Hörnle IA and from Sipplingen–Osthafen are stored at the State Office for Cultural Heritage Baden-Württemberg, Hemmenhofen Office, Fischersteig 9, 78343 Gaienhofen-Hemmenhofen, Germany. The fragments of cereal products from the site of Zürich Parkhaus-Opéra are kept in the permanent repository of the Cantonal Archaeology of Zürich / Kantonsarchäologie Zürich, Stettbachstrasse 7, 8600 Dübendorf, Switzerland.

**Funding:** AGH, FA, HPS, MBA, SMV received funding from the European Research Council (ERC-CoG-2015, GA 682529) https://cordis. europa.eu/project/rcn/202606/ AGH, FA, NB received funding from the Cantonal Archaeology of Zürich http://www.are.zh.ch/internet/baudirektion/ are/de/service/international.html EM received funding from the RBINS https://www. naturalsciences.be/ HK received funding from BRAXAR GmbH http://web.archive.org/web/ 20181108112716/http://brewmaltster.de/ HS received funding from the DFG (62215951) https:// gepris.dfg.de/gepris/projekt/62215951 KMC, LKM, MC received funding from the NCN (UMO-2014/ 13/B/HS3/04976) https://projekty.ncn.gov.pl/index. php?s=7908 LKM received funding from BIAX Consult https://www.biax.nl/ MB received funding from the Japan Society for Promotion of Science (Grant-in-Aid for Scientific Re-search (C), 16K03167) https://kaken.nii.ac.jp/en/grant/kakenhi-project-16K03167/ Additional remarks: Experimental approaches and their evaluation were funded by the European Re-search Council within the framework of the project 'PLANTCULT': Identifying the Food Cultures of Ancient Europe, under the European Union's Horizon 2020 Research and Innovation Program (Grant Agreement No. 682529, Consolidator Grant 2016-2021, PI Soultana Maria Valamoti). Archaeobotanical analysis of the Hierakonpolis material was financially supported by the unit "Quaternary Environ-ments and Humans" of the Royal Belgian Institute for Natural Sciences (RBINS), Brussels. Excavations at Hierakonpolis were undertaken under the auspices of the Hierakonpolis Expedition with funds provided by the Japan Society for Promotion of Science within

installations from Predynastic Egypt, while the three broadly contemporary central European lakeshore settlements lack specific contexts for their cereal-based food remains. The aleurone cell wall thinning known from food technological research and observed in our own experimental material was indeed also recorded in the archaeological finds. The Egyptian materials derive from beer production with certainty, supported by ample contextual and artefactual data. The Neolithic lakeshore settlement finds currently represent the oldest traces of malting in central Europe, while a bowl-shaped bread-like object from Hornstaad–Hörnle possibly even points towards early beer production in central Europe. One major further implication of our study is that the cell wall breakdown in the grain's aleurone layer can be used as a general marker for malting processes with relevance to a wide range of charred archaeological finds of cereal products.

## Introduction

Understanding the role of alcoholic beverages in the evolution of prehistoric societies is fundamental, yet unravelling their patterns of production and consumption is a challenging task. This is especially true for one of the supposedly most ancient and widely spread of these drinks–beer. Beer and other alcoholic beverages play complex roles in human societies [1–5] and, as a consequence, their significance for prehistoric communities has been under discussion for decades [5, 6]. The archaeology of beer is today a highly active field, relating the beverage to the complex social processes involved in the beginnings of agriculture [6–8], to social bonding and stratification in general [2, 9–13], and to the formation of social elites in particular [10, 14–16]. Its ritual and dietary role has been extensively investigated in ancient states with written and iconographic records but our knowledge of the occurrence and manufacture of ancient beer is highly incomplete. Tracking beer in the archaeological record as precisely as possible would therefore result in fundamentally novel insights on human societies in the past. However, archaeological beer finds are still rare and highly contested due to insufficiently explicit criteria, or a missing consensus on the criteria. We present here a discussion of possible markers for beer production from a novel approach: histological evidence observed in charred archaeological crusts and lumps of cereal products as an indication for the beer product itself.

## Defining core processes of beer-making

Beer in its very broadest sense can be characterised as a non-distilled alcoholic beverage produced from a starch-rich source [4, 17, 18]. The transformation of starch as raw material into alcohol as the desired end product requires only two core processes of brewing: (1) the saccharification of starch into mono- and oligosaccharides, and (2) the alcoholic fermentation of the resulting sugars into ethanol [19]. Such a broad definition encompasses beverages as different as South African *kaffir* [20], the *bili bili* [21] in Chad, British porter [22], southern German *Weißbier* [23], Belgian *lambic* [24], and some types of Peruvian *chicha* [25]. Nearly all of the aforementioned, as well as most other known, beer types are based on malted cereals, the growing germ providing the enzymes necessary for starch saccharification [20–26]. Other mechanisms of saccharification utilise the low initial content of endogenous amylases in the unmalted grain, as in *boza* [27] and in *kvass* [28, 29], on the amylases from human saliva as in some *chicha* types [25], or on those extracted from *kōji* fungi as used for the pre-products in *sake* production [30]. Subsequent or synchronous alcoholic fermentation is nearly uniformly

the Grant-in-Aid for Scientific Research (C) programme (proj. no. 16K03167). The analysed materials from Tell el-Farkha were excavated in the 2017 campaign which was funded by the National Science Centre Poland (NCN) as part of the project "Sociopolitical transformations in the Eastern Nile Delta at the transition between the 4th/3rd millennium BC" (grant UMO-2014/13/B/HS3/04976) and which was additionally sponsored by the Jagiellonian University in Kraków, the Archaeological Museum in Poznań, the Polish Centre of Mediterranean Ar-chaeology, the University of Warsaw,and the Patrimonium Foundation, Poznań. The material from Hornstaad—Hörnle IA was unearthed during the 1983–1993 excavations which were funded by the DFG (Deutsche Forschungsgemeinschaft) within the framework of the DFG Schwerpunktprogramm „Siedlungsarchäologische Untersuchungen im Alpenvorland" (PI: Dieter Planck). The finds from Sip-plingen—Osthafen were excavated within the scope of the project "Das 'Sipplinger Dreieck' als Modell jung- und endneolithischer Siedlungs- und Wirtschaftsdynamik am Bodensee" which was also funded by the DFG (proj. no. 62215951, PI: Helmut Schlichtherle). Excavations at Zürich Parkhaus—Opéra were funded by the Cantonal Archaeology of Zürich, the Office for Urbanism of the City of Zürich, and the Federal Office for Culture (FOC) Switzerland, as were the archaeobotanical analyses of fragment ZHOPE 6949.1, carried out at the Vienna Institute for Archaeological Science (VIAS) at the University of Vienna in 2014. The State Office for Cultural Heritage Baden-Württemberg and the Institute for Botany (210) of the University of Hohenheim funded the international workshop "Ancient beer: multidiscipli-nary approaches for its identification in the archaeological record" held at the University of Hohenheim in February 2019, during which the foundations for this paper were laid. The comparative find no. 252 from Haselbach was obtained from the project „Keltische Siedlungszentren in Ostösterreich" (PI: Peter Trebsche and Stephan Fichtl) funded by the Federal Government of Lower Austria. Funders BIAX Consult and Braxar GmbH provided support in the form of salaries for authors LKM and HK, respectively, but did not have any additional role in the study design, data collection and analysis, decision to publish, or preparation of the manuscript. The specific roles of these authors are articulated in the 'author contributions' section. Neither had the other funders a role in study design, data collection and analysis, decision to publish, or preparation of the manuscript.

**Competing interests:** I have read the journal's policy and the authors of this manuscript have the

based on the metabolism of true yeasts (order Saccharomycetales), accompanied to varying extents by the action of lactic acid bacteria (order Lactobacillales, LAB) and/or acetic acid bacteria (family Acetobacteraceae, AAB), which also feed on the saccharified starch but do not produce ethanol.

The two core processes of starch and (hemi-)cellulose saccharification and subsequent alcoholic fermentation require only two additional actions, or processes, as prerequisites: an initial soaking with water leading to germination, and the crushing or grinding of the malted grains in order to break down cell walls and enlarge sufficiently the surfaces exposed to the fermenting microorganisms. Other processes can significantly enhance saccharification and fermentation, such as alternating soaking/drying phases and various temperature regimes, with purposes as different as 1) fostering sprouting, 2) killing the germ, 3) gelatinizing and thus solubilizing the starch, or 4) optimizing the enzymatic processes during mashing [31, 32]. However, although some of the aforementioned have already been postulated for prehistory [33, 34], they are not at all necessary for successfully brewing a drinkable beer [35]. It may therefore be more useful to postulate a *chaîne opératoire* for brewing which opposes core actions and processes (Fig 1) to some major optional ones.

## Current archaeological approaches towards ancient brewing

There is ample archaeological and epigraphic evidence for the installations, implements and drinking vessels associated with beer production and consumption. These data inform on brewing practices as well as on feasting on beer [2, 4, 9–13, 33].

Biogenic sources of information–at least potentially–give more or less direct evidence on the raw materials used, on the processes involved in their transformation and on the final product itself (Fig 1) [4, 37, 38, 42, 43], even when no clear epigraphic and archaeological evidence is preserved. While the mere presence of cereal grains in an archaeological site only allows for speculations on the method of their intended consumption, it is now commonly accepted [37] that large quantities of evenly-sprouted grains can be regarded as the results of intentional malting. They are therefore interpreted as indicators of beer making. Such malt finds are indeed not so rarely preserved in a charred state [44–52], while finds of desiccated sprouted grains are limited to arid environments [42, 53, 54].

As soon as these sprouted and possibly kilned grains are crushed or ground, the morphological traits of malt as an identifiable germ are lost and the obtainable evidence becomes less clear [42, 54–56]. Other processes such as saccharification and fermentation are difficult to track as their evidence does either not always preserve or it leaves too much interpretational ambiguity. Ethanol as the *conditio sine qua non* of brewing would certainly make an excellent biomarker if it was only preserved in archaeological contexts. The carbon dioxide produced in the fermentation process has only very little chance to be retained in a liquid and to be observed in an archaeological deposit–very unlike doughs, which retain and preserve the gas bubbles as cavities [57, 58]. Neither do other primary metabolites directly linked to fermentation, such as lactic acid or acetic acid, survive the centuries. Recent approaches have therefore focussed on finding new indicators of ancient brewing using plant histological and chemical markers: Under the SEM, starch granules from desiccated beer residues in Egypt [56, 59] have successfully delivered the degradation traces typical of amylase attack, thus serving as clear indicators of malting. The same desiccated materials have delivered direct evidence of yeast cells [56]. Calcium oxalate crystals which can form during mashing and fermenting [31, 40] (Fig 1) have also been suggested as markers for brewing [7, 60], even if they have recently been challenged due to their lack of diagnostic value given the ubiquity of calcium oxalate in the plant kingdom [4, 35, 37, 61]. Approaches interpreting singular damaged starch granules as

following competing interests: LKM and HK are
paid by commercial companies (BIAX Consult and
Braxar GmbH, respectively). This does not alter
their adherence to PLOS ONE policies on sharing
data and materials.

evidence for early brewing [34, 59] have also already met critical opposition [8]. Recent work in proteomics seems to be highly promising [62] but will still require thorough evaluation in the coming years. Due to this overall complicated situation, ancient beer research is constantly searching for new tools, involving new and more reliable markers.

## A new look at charred finds of cereal products

Within the scope of the ERC project PLANTCULT, systematic microscopical analyses not only of entire charred "bread buns"–bread-like objects, to be more precise [58]–but also of amorphous charred objects (ACO) from a wide range of mainly prehistoric sites across central and south-eastern Europe were carried out [58, 63–66], and some finds from other regions such as Egypt were re-evaluated. The overall goal was to gather diagnostic characters of the components and operational sequences involved in the production of the cereal products concerned, and eventually to establish systematised approaches for the analysis and classification of this find category [58, 63, 67].

Although occurring regularly in archaeological contexts, amorphous fragments of charred cereal products are a particularly difficult material to approach [67], not least because of their genesis. Grinding destroys a grain's morphology, and as a consequence also the aforementioned traces of possible sprouting or malting. Furthermore, the material is also heavily affected by massive chemical transformations during pyrolysis [68, 69], which limits chemical analyses of such materials to mostly imprecise results [70–76]. However difficult the situation may still be, interpretations of ACO containing ground cereal remains have still been able to approach a wide range of possible foodstuffs, ranging from porridge-like finds to others resembling pre-cooked bulgur, to air-dried cereal preparations similar to pasta and *trahanas*/*tarhana* and also to bread and beer in the widest sense [47, 63, 66, 67, 75, 77–83].

The current study brings forward a new histological feature which can be used for the detection of malted grains in charred lumps or crusts of cereal products, even if the components are finely ground or no intact starch granules are preserved. The precise identification of malt-based foodstuffs enables this feature to serve as an additional indirect marker in the field of "ancient beer" research, entirely independent of other previously proposed markers.

## Introducing aleurone cell wall degradation as a new marker for malting

**Characteristics of the aleurone cell wall.** The aleurone layer is a secretory tissue forming the outermost endosperm layer of the grains of the grass family (Poaceae). In contrast to the inner starchy endosperm, it remains a living tissue in the mature grain during caryopsis development [84], with the aleurone cells storing a significant portion of the grain's protein content. In wild grasses as well as in nearly all cultivated cereals, the aleurone layer is typically a single cell thick (Fig 2), with the known exception of domesticated barley (*Hordeum vulgare*) which regularly develops multi-layered aleurone [85, 86]. Albeit only a primary cell wall, the aleurone wall is rather massive, usually at least twice as thick as in the central endosperm cells reported for numerous Old World grasses [86, 87]. Measurement means of double cell wall thickness in air-dried material centre around 3 μm [88] in barley (*Hordeum vulgare*) and range from 2.5 μm [89] to 4.9 μm [90] in bread wheat (*Triticum aestivum*).

This cell wall is mainly composed of hemicelluloses. For *Triticum aestivum*, these are reported as consisting roughly one third (1,3;1,4)-β-glucan and roughly two thirds arabinoxylan (AX) [91, 92], the latter incorporating phenolic compounds such as ferulic acid [93]. The β-glucan: AX ratio as well as the arabinose: xylose ratio of the AX can vary significantly between species and even between cultivars, with environmental factors apparently also having

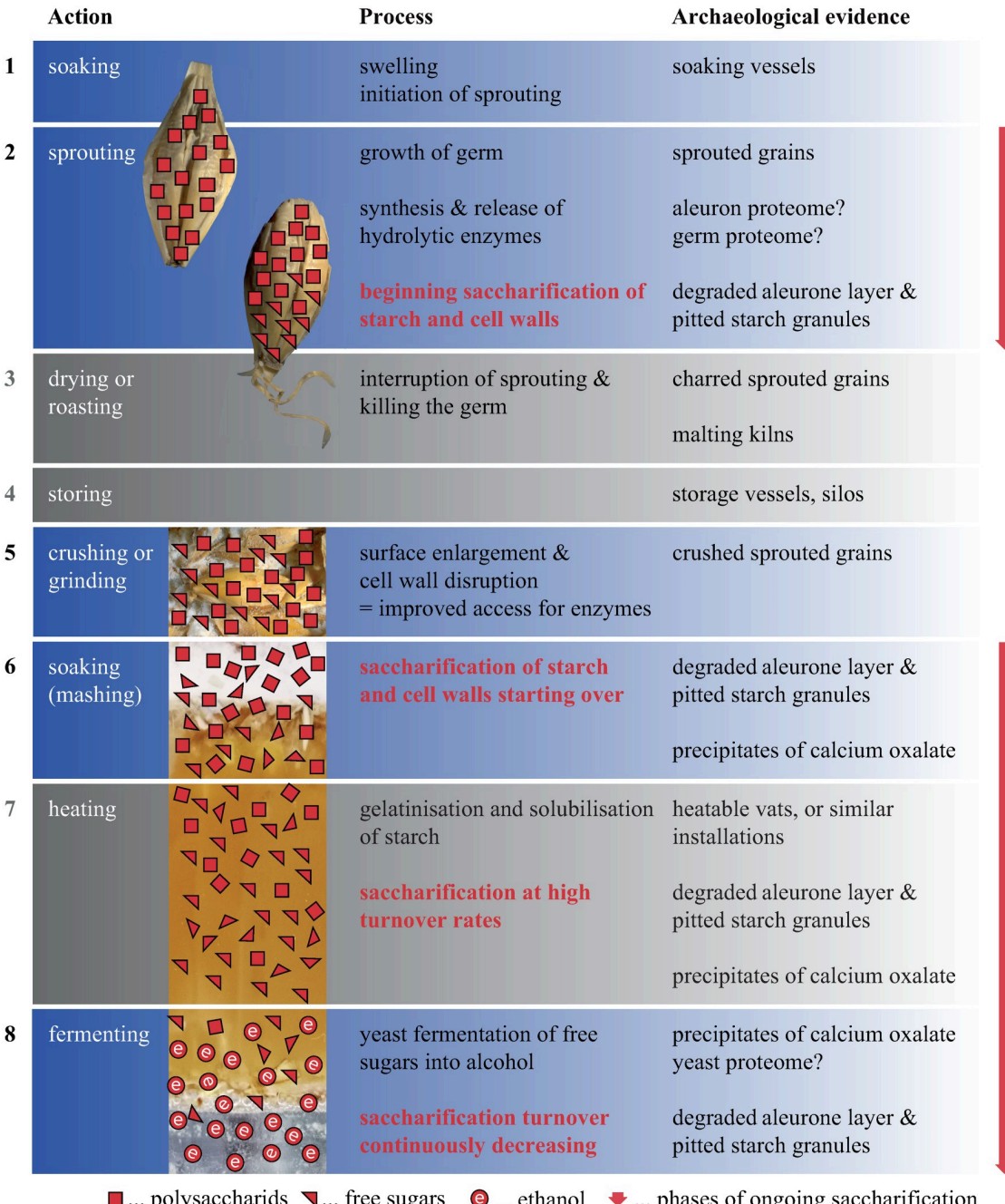

| | Action | Process | Archaeological evidence |
|---|---|---|---|
| 1 | soaking | swelling<br>initiation of sprouting | soaking vessels |
| 2 | sprouting | growth of germ | sprouted grains |
| | | synthesis & release of hydrolytic enzymes | aleuron proteome?<br>germ proteome? |
| | | **beginning saccharification of starch and cell walls** | degraded aleurone layer & pitted starch granules |
| 3 | drying or roasting | interruption of sprouting & killing the germ | charred sprouted grains<br><br>malting kilns |
| 4 | storing | | storage vessels, silos |
| 5 | crushing or grinding | surface enlargement & cell wall disruption = improved access for enzymes | crushed sprouted grains |
| 6 | soaking (mashing) | **saccharification of starch and cell walls starting over** | degraded aleurone layer & pitted starch granules<br><br>precipitates of calcium oxalate |
| 7 | heating | gelatinisation and solubilisation of starch | heatable vats, or similar installations |
| | | **saccharification at high turnover rates** | degraded aleurone layer & pitted starch granules<br><br>precipitates of calcium oxalate |
| 8 | fermenting | yeast fermentation of free sugars into alcohol | precipitates of calcium oxalate<br>yeast proteome? |
| | | **saccharification turnover continuously decreasing** | degraded aleurone layer & pitted starch granules |

■ ... polysaccharids  ◣ ... free sugars  ⓔ ... ethanol  ⬇ ... phases of ongoing saccharification

**Fig 1. Simplified *chaîne opératoire* of brewing actions together with their associated processes and traces in the archaeological record.** The diagram is based on historical and ethnographic records as well as on knowledge of modern brewing technology [4, 31, 32, 36–40]. Only processes required for the formation of alcohol are described. Blue background: "core" actions and processes, grey background: optional actions and processes which are, for example, characteristic of modern beer making [19, 41]. Saccharification takes place in numerous stages of beer making, which is additionally illustrated by red arrows. Figure: University of Hohenheim / M. Berihuete Azorín; Office for Urbanism Zürich / N. Bleicher; ÖAW-ÖAI / A. G. Heiss; TUM-Weihenstephan / M. Zarnkow.

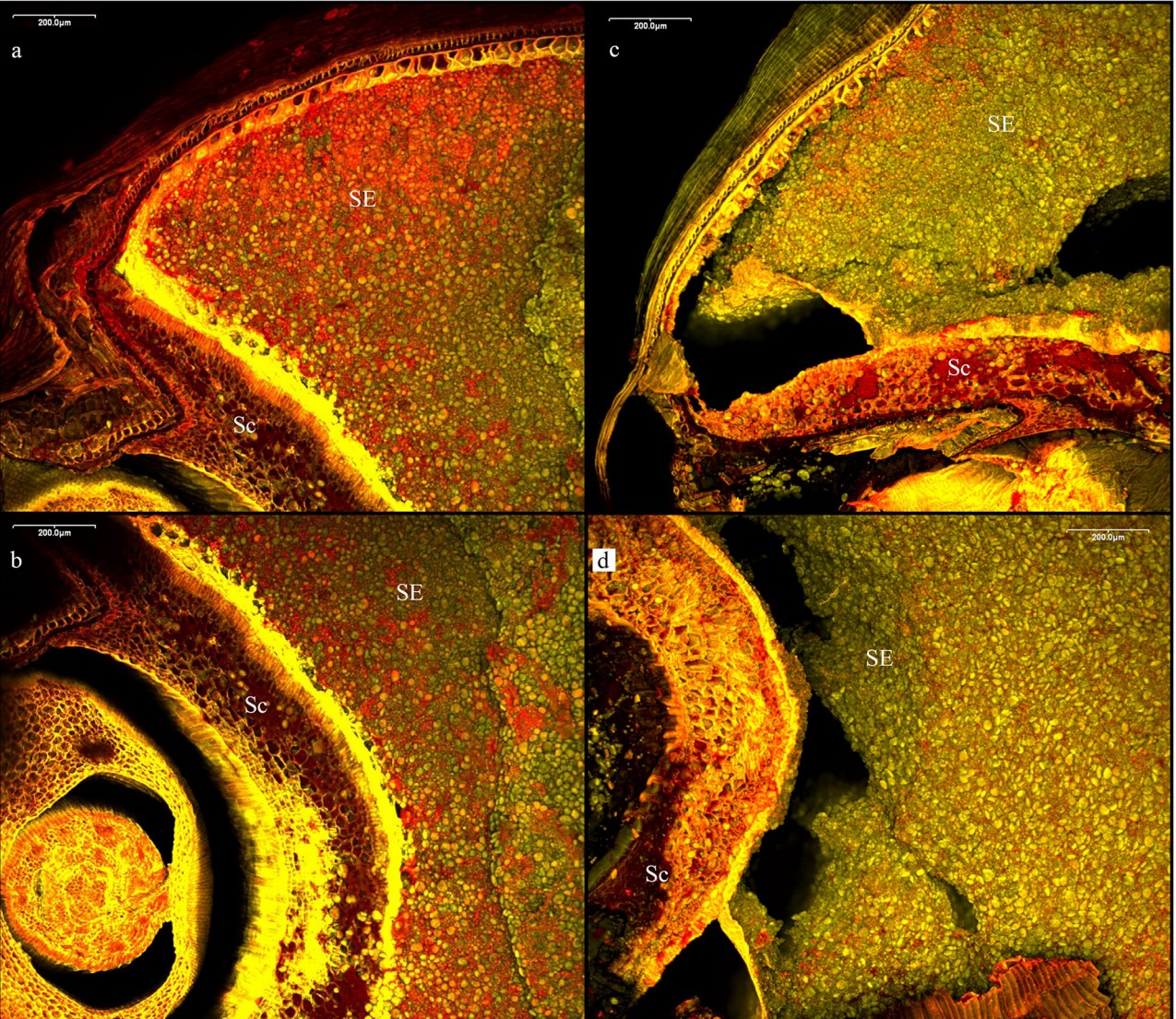

**Fig 2. Embryo end of a bread wheat (*Triticum aestivum*) grain showing massive degradation of the endosperm.** a, b) unmalted grain, and c, d) after six days of sprouting. SE. . . starchy endosperm, Sc. . . scutellum. Fluorescence (CLSM) colours: yellow = cell walls, red = proteins, green = starch. Image: TUM-Weihenstephan / J. Helbing.

a strong influence on hemicellulose composition [94–97]. In all cases, cellulose contributes a minor component of the aleurone's cell walls [98].

**Cell wall degradation during sprouting.** Germination–the key process of malting–is a unique process in food preparation as it is entirely endogenous, needing an intact, living grain synthesising the required enzymes for saccharification of the mash, as these hydrolases are virtually absent in the dormant grain. Moderated by the release of gibberellic acid (GA) by the embryo after soaking [99–103], they are synthesised *de novo* during germination. Starting at the embryo end of the grain [104], α-amylase and numerous other hydrolytic enzymes are synthesised in the scutellum and in the aleurone layer, including glucan 1,4-β-glucosidases (β-D-glucan glucohydrolases) and endoxylanases, which lead to progressive cell wall breakdown [99, 102, 105–107]. The overall enzyme release reaches its main peak after approx. 24 h [102,

106]. This enzymatic attack on cell walls not only provides additional carbohydrates for the growing embryo but also enables permeation of the amylases throughout the endosperm [104] (Fig 2). Other processes potentially influencing aleurone cell wall thickness are discussed and critically evaluated in the Discussion.

**A diagnostic feature hidden in plain sight?** One of the most significant structural changes during the germination of a grain is the breakdown of cell walls as a characteristic and recurrent feature, making degraded cell walls an indicator of malted grains on a microstructural level, even if the grain is ground into grist.

While the breakdown of the inner endosperm cell walls is crucial for the success of brewing, the thinning of aleurone cell walls is not. It is nonetheless easily observable [108]. The breakdown of aleurone cell walls *in vitro* has been described as "extensive" already 12 h after exposure to gibberellic acid [109], while measurable *in vivo* losses in dry weight of cell wall fractions (aleurone and testa) in barley have been reported to reach up to 42% after 2 days of germination and up to 58% after 6 days [110] (Figs 2 and 3), accompanied by the formation of intercellular spaces [99] and the progressive apoptosis of the aleurone cells [109, 111]. While endosperm cell walls have been observed to be entirely digested after 12 days of germination, aleurone cell walls can still remain intact, albeit massively thinned [112].

The process of aleurone activation and general cell wall breakdown is, however, not evenly distributed along the cereal grain. Furthermore, diffusion of the hydrolytic enzymes within the still intact tissue is much slower than *in vitro*. While on day six of germination the thinning of the aleurone cell walls is clearly observable under the microscope close to the embryo, the distal end remains largely unmodified at the same time [113].

It may be due to the lack of practical relevance for modern brewing that these observations on aleurone cell wall degradation have not received much attention beyond plant physiological and brewing technology research. Neither has this knowledge ever been transferred to the archaeological sciences. Aleurone cell wall thinning could definitely be a very promising marker for malting and, while only indirectly pointing towards beer, might prove highly useful for researchers of ancient beer. Not for all types of beer, as must be admitted. Beer types entirely basing on unmalted grains cannot be detected via aleurone cell wall thinning, because microbial fermentation itself has no impact on cell wall thickness: Wild types of true yeasts do not display notable cellulase or xylanase activity [24, 114], and neither do wild LAB or AAB strains [115–119]. A few

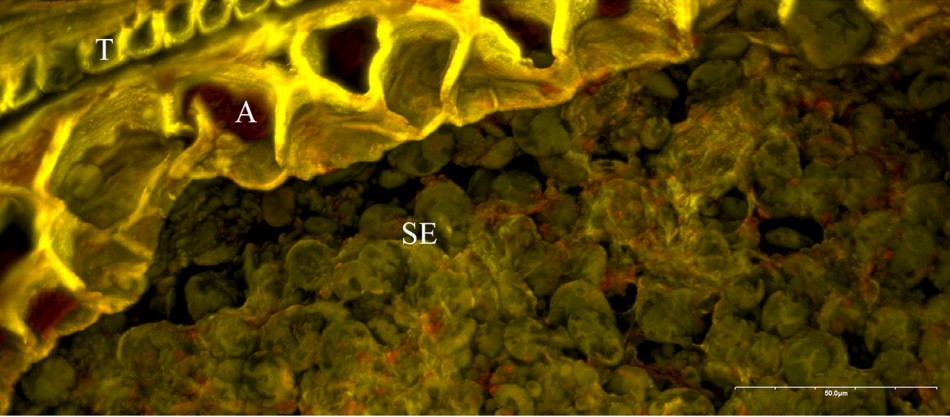

**Fig 3. Bread wheat (*Triticum aestivum*) grain after six days of sprouting.** SE. . . starchy endosperm, A. . . aleurone. T. . . transverse cells. Fluorescence (CLSM) colours: yellow = cell walls, red = proteins, green = starch. Image: TUM-Weihenstephan / J. Helbing.

species in the genera *Enterococcus* and *Pediococcus*–mainly intestinal bacteria [120, 121]–do indeed exhibit cellulase activity, yet at rates far too low for being relevant in the time-scale of beer production [117]. Aside from beer, we want to stress that a marker for possible archaeological malt remains also unlocks research into a far large variety of other malt-based foodstuffs and beyond (see Discussion).

## Research goals

Archaeobotanical studies of grain tissues have already successfully contributed to the knowledge of the ingredients and processes involved in the production of cereal-based foodstuffs, not only of entire "loaves" [58, 66, 122] but also of small fragments [75, 79, 80, 82, 123, 124]. These studies have demonstrated that patches of aleurone tissue are not only still recognizable in charred fragments of ground food preservations from archaeological contexts (Fig 4), but the aleurone tissue's structure [86, 87] has also successfully been used for the differentiation of barley (*Hordeum vulgare*) from other Old World cereals and grasses [58, 66, 80, 82, 122, 125, 126].

As laid out above, the phenomenon of cell wall degradation in cereal grain aleurone as well as the mechanisms behind it (i.e. grain germination) are well-known, and are well-documented by ample literature (see previous section). Instead of replicating these experiments, our research goals were rather: 1) the detection of recognizable aleurone affected by sprouting (i.e. with degraded cell walls) in charred cereal grains, and 2) the critical evaluation of whether similar structural changes in archaeological food remains could indeed be a suitable indicator for malting in the archaeological record.

**Experimentally charred malt.** We used commercially available barley malt in different stages of sprouting, which was then experimentally charred (see Methods section), in order to test whether the conspicuously thinned cell walls in the grain's aleurone layer are still observable after charring (Figs 5 and 6). We did not consider unmalted grain in our sequences, as it is known from the literature that no *in vivo* changes occur in the first 24 hours of sprouting [113]–particularly not in the grain's middle section which we analysed (see below). We were able to confirm this by comparing the measurements on experimental material (see Results section).

**Archaeobotanical case studies.** In Egypt, some insights into Predynastic brewing have already been obtained from three sites–Abydos, Hierakonpolis, and Tell el-Farkha. Excavations at all three sites have uncovered installations that may well be connected with brewing, involving actual brewing vat contents. At Abydos, the residues contain wheat, but presumably emmer (*Triticum dicoccum*) [42] instead of "common wheat" as originally stated [127]. Two further sites, both dating to the New Kingdom, have provided important evidence for the sequence of the ancient brewing method: Deir el-Medina [128] and the Workmen's Village at Amarna [42]. In our study, charred crusts and lumps of emmer-based cereal products retrieved from the 4th millennium BCE brewing installations in Hierakonpolis and Tell el-Farkha (Fig 7) were re-evaluated in order to test the consistency and reliability of thinned aleurone cell walls as indicators of ancient malting.

From the central European cereal products, specimens exhibiting conspicuously thin aleurone cell walls were pre-selected and compared to the experimental findings as well as to the Egyptian brewing remains. These charred processed food remains from central European Late Neolithic (4th millennium BCE) lakeshore settlements (Fig 7) situated at Lake Constance (southwestern Germany) and Lake Zürich (Switzerland) had not been previously connected to specific culinary practices such as brewing activities and/or dough preparation.

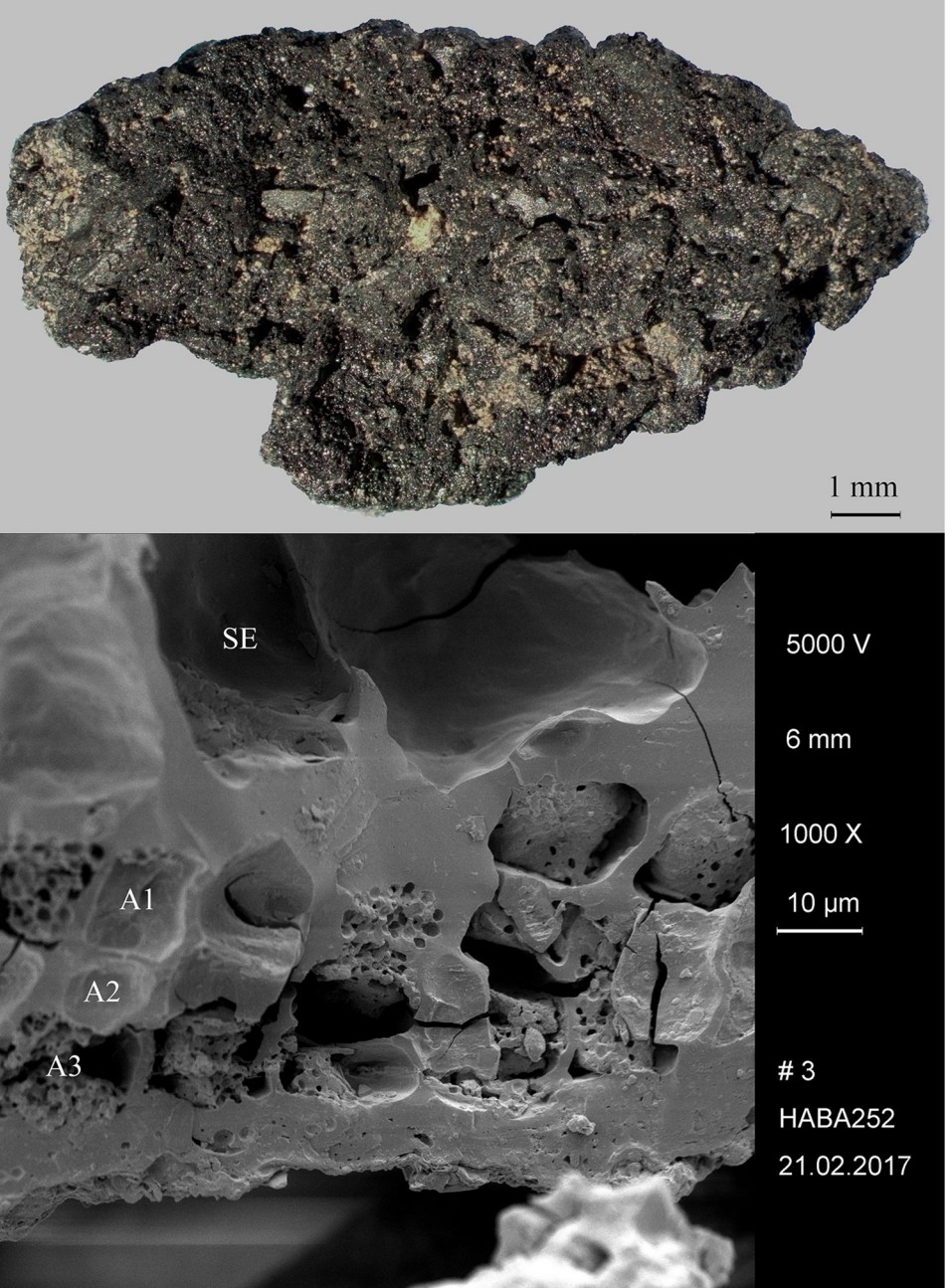

**Fig 4. Archaeological barley (*Hordeum vulgare*) aleurone with regular wall thickness.** Fragment of a charred cereal product from the La Tène C period (c. 250–150 BCE) site of Haselbach, Lower Austria (find no. 252, SE 16–03 = SE 16–19). Top: light micrograph, bottom: SEM image. SE. . . starchy endosperm (fused remains), A1–A3. . . aleurone layers. Image: ÖAW-ÖAI / A. G. Heiss.

## Materials and methods

### Ethics statement

No permits were required for the described study, which complied with all relevant regulations.

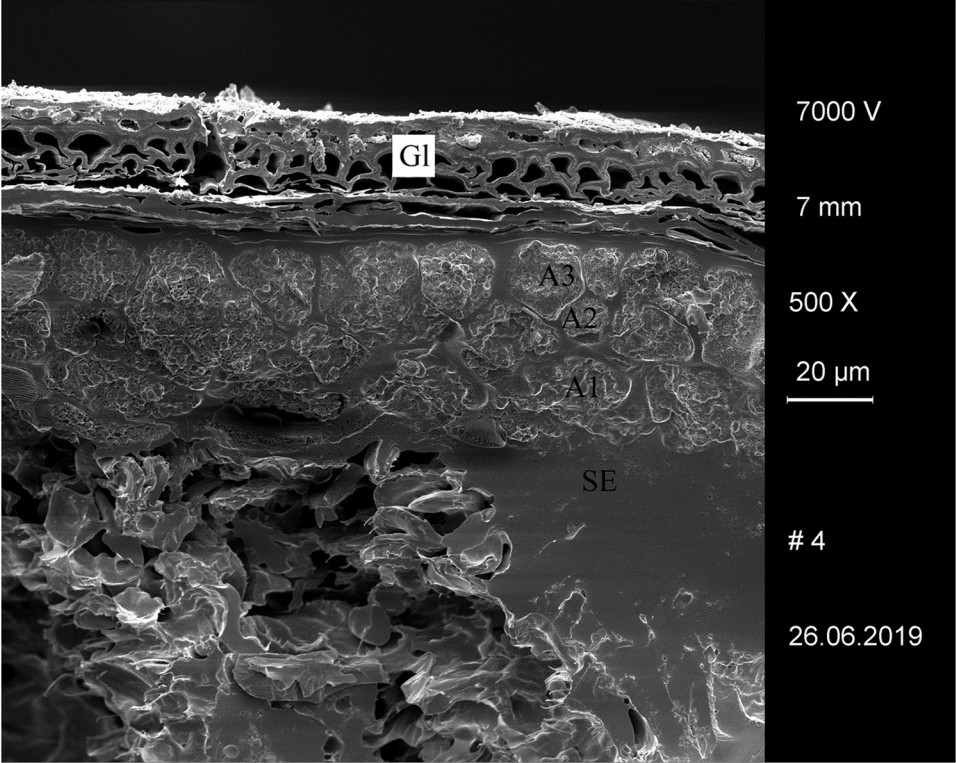

**Fig 5. Experimentally charred hulled barley (*Hordeum vulgare*) grain after 1 day of malting.** Gl. . . glume, A1–A3. . . aleurone layers, SE. . . starchy endosperm (fused remains). Image: University of Hohenheim / M. Berihuete Azorín.

## Data repositories and location of specimens

All relevant data are contained within the manuscript and its Supporting Information files. The experimentally charred barley malt is accessible for scientific re-evaluation on request at the University of Hohenheim, Department of Molecular Botany (190a), Garbenstraße 30, 70593 Stuttgart, Germany. All archaeological samples are accessible as well for scientific re-evaluation on request: The brewing residues from Hierakonpolis are stored at the magazines at the site and can be accessed by obtaining permission from the Egyptian Antiquity Authorities. Likewise, the materials from Tell el-Farkha are accessible for re-evaluation after the aforementioned permissions. The charred cereal products from Hornstaad–Hörnle IA and from Sipplingen–Osthafen are stored at the State Office for Cultural Heritage Baden-Württemberg, Hemmenhofen Office, Fischersteig 9, 78343 Gaienhofen-Hemmenhofen, Germany. The fragments of cereal products from the site of Zürich Parkhaus-Opéra are kept in the permanent repository of the Cantonal Archaeology of Zürich / Kantonsarchäologie Zürich, Stettbachstrasse 7, 8600 Dübendorf, Switzerland. No permits were required for the described study, which complied with all relevant regulations.

## Reference material used for illustrating the effects of malting

One part of bread wheat (*Triticum aestivum*) grains harvested in 2007 was malted according to the MEBAK standard methodology for malting hulled barley [129], while the other part was

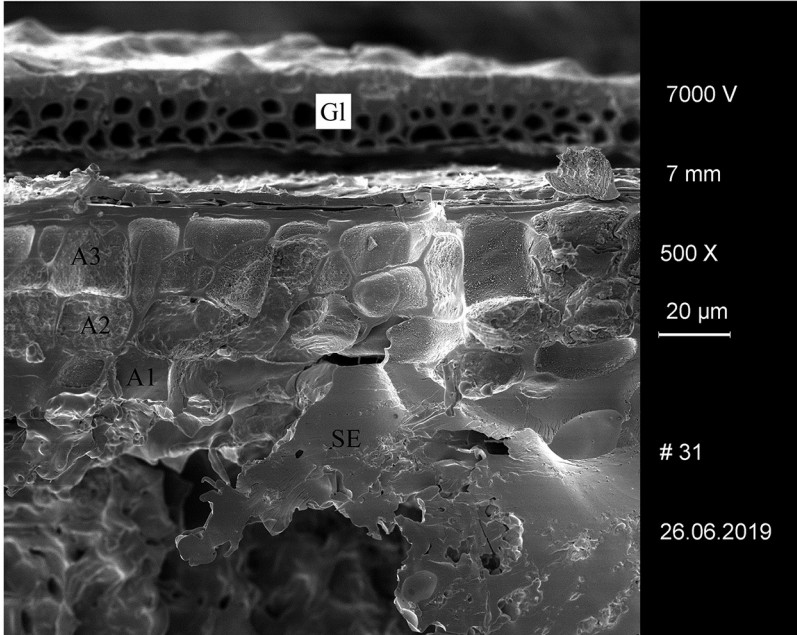

**Fig 6. Experimentally charred hulled barley (*Hordeum vulgare*) grain after 5 days of malting.** Gl. . . glume, A1–A3. . . aleurone layers, SE. . . starchy endosperm (fused remains). Image: University of Hohenheim / M. Berihuete Azorín.

kept unchanged as reference. All sample grains, malted and unmalted, were deep-frozen at -20 ˚C and embedded into wax. Sections were made by hand using a razor blade [130]. Differential fluorescence staining for cell wall components (Calcofluor White), starch (FITC), and proteins (rhodamine B) was applied onto the sections [130]. Images of the specimens were produced while they were still in a wet state using an Olympus FV300 Confocal Laser Scanning Microscope (CLSM), equipped with a diode laser (405 nm), an argon laser (458 nm) and a helium neon laser (543 nm). Image stacks of 50–100 individual images (0.2–5 μm steps each) were then combined into the final images with enhanced depth-of-field using the software Olympus Fluoview.

## Experimentally charred barley malt

Sprouted two-row hulled barley (*Hordeum vulgare* subsp. *vulgare* f. *distichon*) was provided by Heinrich Durst Malzfabriken, Bruchsal-Heidelsheim, Germany, in various states of germination (1 day, 2 days, 3, days, 4 days, 5 days). Charring is well-known for its non-linear effects on the dimensions of various plant seeds [131–141] yet in recent experiments we were able to show that cereal grains do maintain proportions very close to the uncharred state when charring at low temperatures [142]. A certain shrinkage does, however, always occur due to the loss of water and other volatile substances [68, 69, 143, 144]. Charring at higher temperatures, however, affects overall grain morphology in a more drastic way, while temperatures exceeding 300 ˚C will even lead to a loss in mass due to the thermal degradation of hemicelluloses and cellulose with reported shrinkage rates of typically 30% and beyond [145, 146]. The possible implications of this phenomenon will be elaborated later (see Discussion). We decided to use low-temperature regimes and char all observed grains under the same conditions. The barley

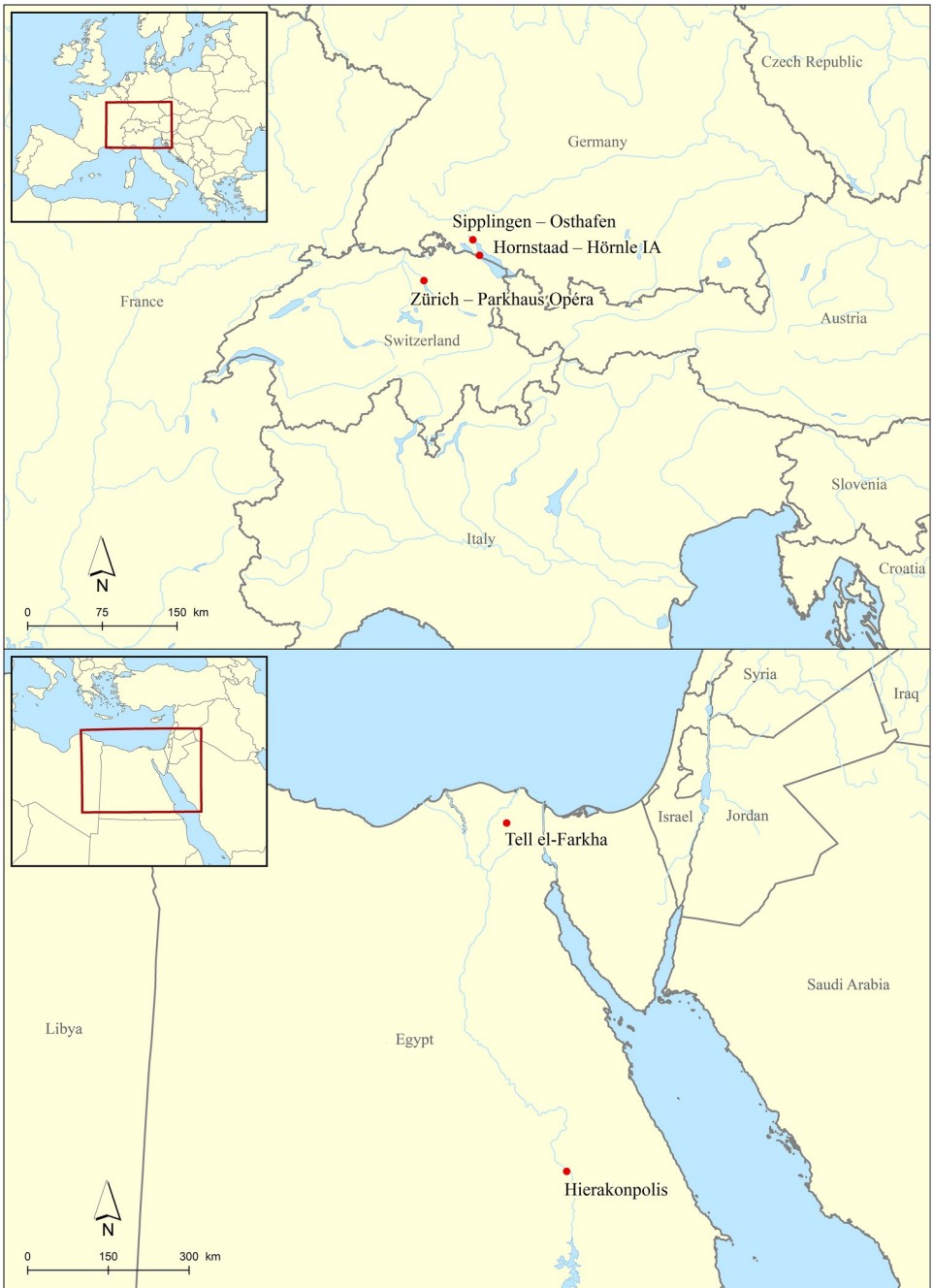

**Fig 7. Site locations of the archaeological case studies.** Top: central European sites, bottom: Egyptian sites. Map: ÖAW-ÖAI / C. Kurtze, A. G. Heiss.

malt used for this experiment was first dried in an electric drier for 24 hours and charred afterwards in a Nabertherm NA 15/65 muffle furnace at low temperature (230 ˚C) for 24 hours [142]. The charred grains were broken in half in order to observe their middle sections. SEM imagery was produced using a Zeiss DSM 940 after sputter coating with gold/palladium in a Balzers SCD 040.

## Brewing remains from predynastic Egypt (4<sup>th</sup> mill. BCE)

**Hierakonpolis—HK 11C.**   Located near the modern town of Edfu, Hierakonpolis was the major Predynastic political centre in Upper Egypt. Excavations at locality HK 11C revealed a well-preserved brewery consisting of eight freestanding ceramic vats, with diameters ranging from 60 to 85 cm [147]. The vat exteriors had been coated with mud and pottery sherds for protection from thermal shock and the promotion of even heating. A ring of large sherds cemented together with mud placed around the base of each vat aided stability and enclosed the fire. Adhering to the interior of five vats was a thick layer (up to 3 cm) of shiny black residue. Its archaeobotanical examination indicated that the main ingredient was emmer wheat [148]. These grains were coarsely crushed or ground and then heated in water together with grains that supposedly had been malted in their husks. The SEM revealed the presence of pitted starch grains indicative of amylase attack from the saccharification process (Fig 1). Radiocarbon dating of the residues (raw date: $^{14}$C 4875±40 BP [147]) puts them in the time range of 3764–3537 cal. BCE (2 sigma), making this installation the oldest dated brewery in Egypt (Naqada IC–IIB). A fragment of the residues from Vat 06 (Fig 8) was subjected to in-depth archaeobotanical and chemical analysis.

**Tell el-Farkha—Western kom.**   Tell el- Farkha is located on a *gezira* (sandy island) in the Eastern Delta, approximately 120 km northeast of Cairo. The site is marked by three mounds, designated as Western, Central, and Eastern Kom. At the time of the Lower Egyptian culture (Naqada IIB), until the middle of the First Dynasty, Tell el-Farkha was one of the most important towns (perhaps even a capital) and administrative-cultic centre of the Eastern Delta. In the middle of the First Dynasty, Tell el-Farkha changed its role–from the capital to a provincial town of only economic significance [149]. A complex of five successive breweries dated to the Lower Egyptian Culture was discovered on Western Kom.

The oldest structure (known as Feature 192/201) is archaeologically dated to the Naqada IIB–C period (3600–3500 BCE). It was arranged as parallel rows of vat sets, and it measured 6 meters by 3.4 meters. There were 13 circular installations (or sockets) in which the vats were originally seated and their contents were heated. Two large ceramic vats were preserved *in situ*. One of the vats contained charred organic residue from the cooking or heating of the grain, which was subjected to archaeobotanical and chemical analysis.

A striking feature of the residue lumps was the combined presence of a completely fused matrix and coarsely processed (most likely crushed) emmer grains [150]. The SEM revealed that the fused matrix is also made of emmer grains. Areas of a single-celled aleurone layer were the most common type of particles observed in the matrix (Fig 9). There was noticeably very little chaff. No morphologically identifiable starch granules were found in the grain endosperm, pointing towards either the starch content's complete gelatinisation or its complete saccharification. The diversity of residue structure, the combination of fused emmer grain matrix and the coarsely broken grains suggest two differently-treated portions of emmer grain, which at a certain point were mixed together. The first portion would have been well-cooked in water, with the later addition of the second portion of coarsely broken, uncooked grain. Besides the vat contents, there was also mineralized organic material found within the brewery complex. Some of these mineralized masses contained large quantities of coarsely shredded chaff, most probably representing a by-product such as spent grain that had been removed from the liquid (possibly by being sieved out) and discarded. Mineralised lumps of fine-grained organic masses were also found within the brewery installations. Embedded into their matrices were only a few small fragments of chaff, together with numerous small cavities. These mineralised lumps are currently interpreted as direct evidence of the fermentation process itself–the sludge that had deposited at the bottom of the fermentation vessels, its cavities originating from the carbon dioxide generated by the yeast and other fermenting microorganisms.

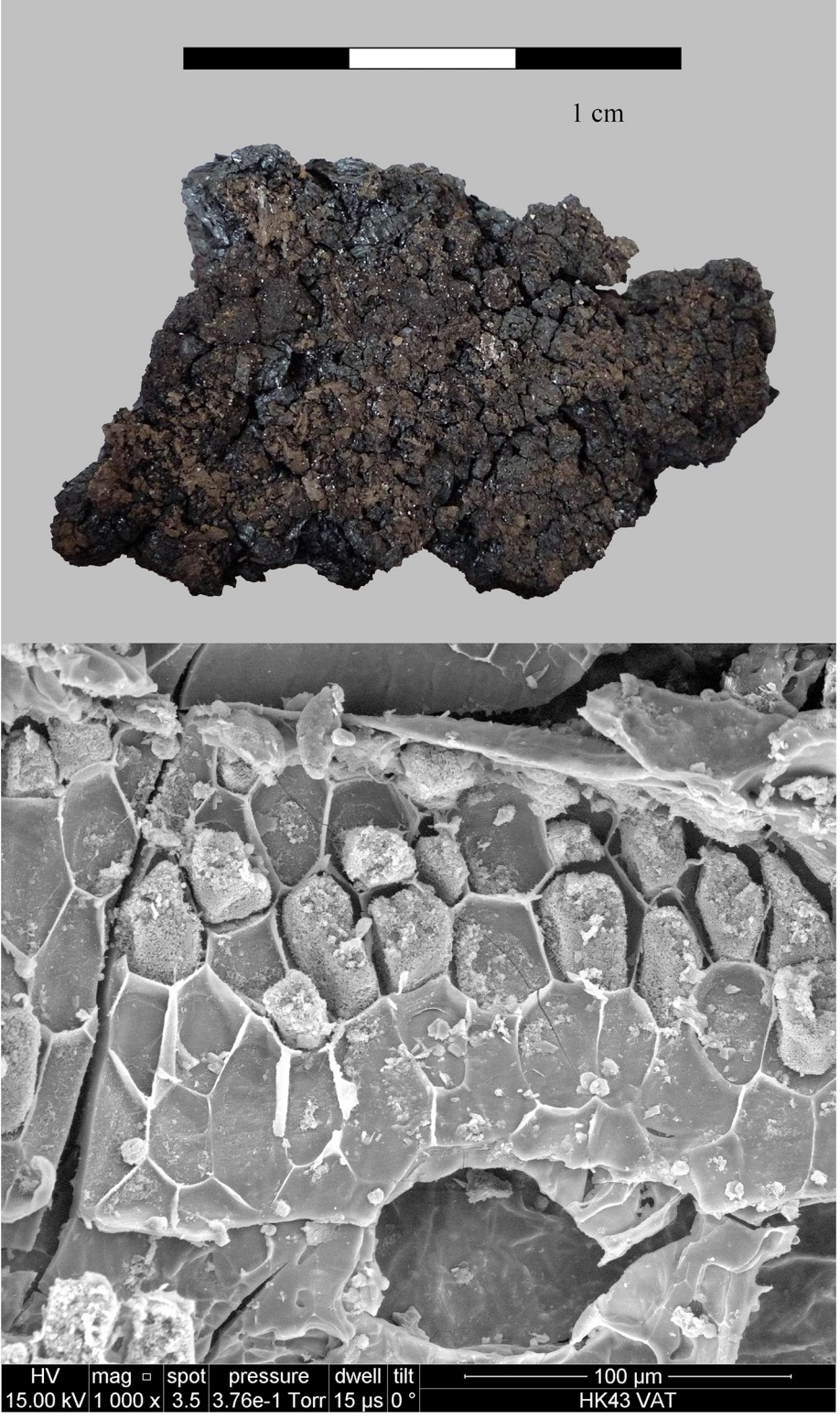

**Fig 8. The partially charred cereal product from Hierakonpolis.** Find no. HK 11C. Top: light micrograph, bottom: SEM image. The material has been previously identified as emmer (*Triticum dicoccum*) based on caryopsis macroremains [148] in the food crust. Image: Helwan University, Cairo / E. A. E. Attia.

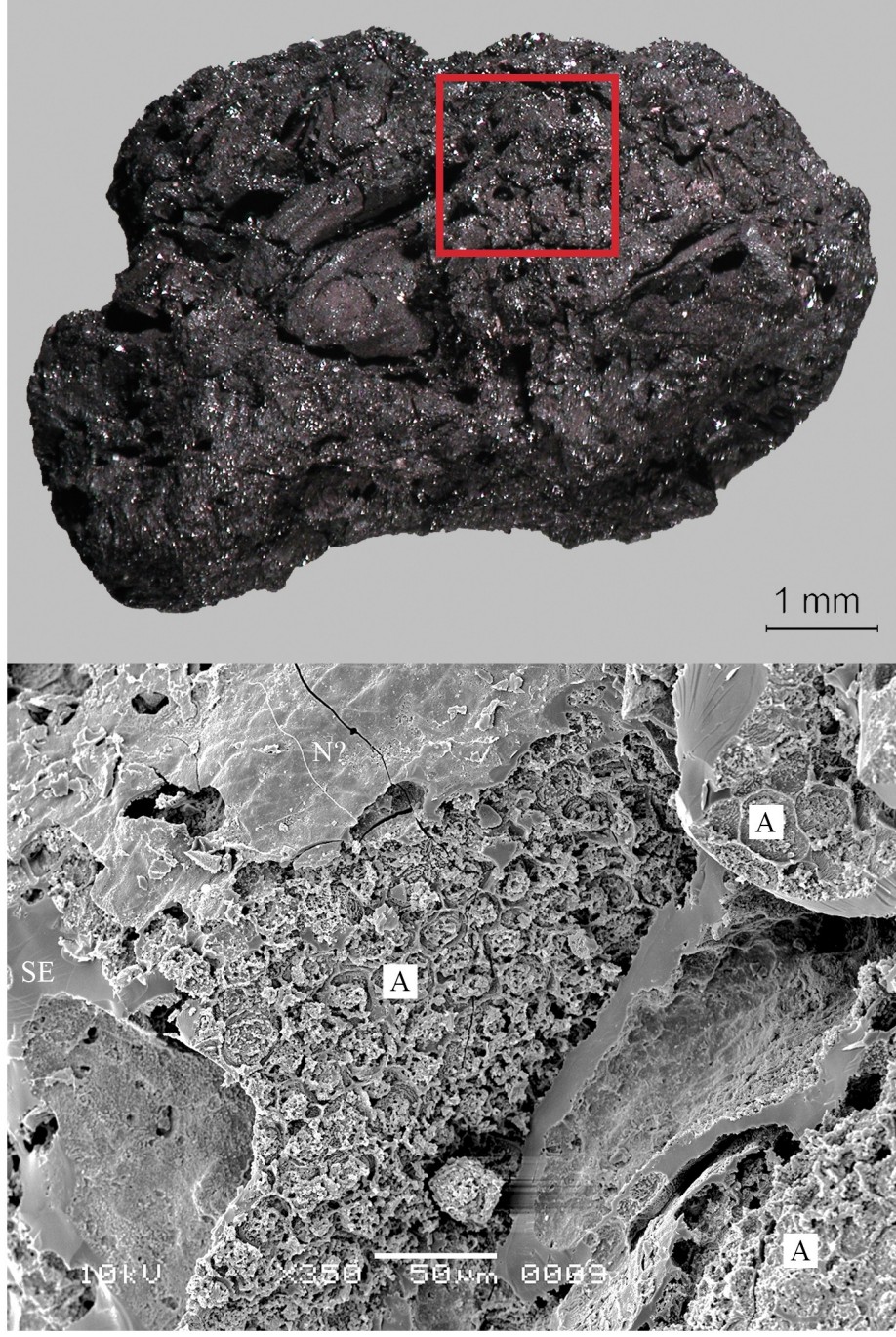

**Fig 9. The charred emmer (*Triticum dicoccum*) product from Tell el-Farkha.** Feature 192/201 (vat contents). Top: light micrograph (red square: location of SEM subsample), bottom: SEM image. A. . . three patches of single-layered aleurone, SE . . . starchy endosperm (fused remains), N?. . . nucellus tissue. Image: Naturalis Biodiversity Center, Leiden / L. Kubiak-Martens.

## Amorphous charred objects (cereal preparations) from late neolithic central Europe (4<sup>th</sup> mill. BCE)

**Hornstaad—Hörnle IA.** This Late Neolithic lakeshore settlement is situated on the bank of Lake Constance's smaller zone–the *Untersee* in southwestern Germany. Due to waterlogged

preservation conditions, the site yielded extraordinarily large amounts of well-preserved plant remains, which have been studied in detail [151]. Archaeological analyses resulted in the interpretation that the houses were erected in the shallow water zone [152]. The cultural layers mostly represent everyday refuse and remains of the buildings. The massive burnt layer AH2 from the Pfyn culture, most probably the result of a catastrophic fire, was dendrochronologically dated to 3910 BCE [153]. A cup-shaped cereal product (find no. Ho 45/43-28, Fig 10) from this destruction layer was re-examined in the current study. One of the object's halves was documented via photogrammetry (see supplementary information) before subsampling.

**Sipplingen—Osthafen.** Also situated on Lake Constance, this late Neolithic lakeshore settlement likewise displayed excellent waterlogged preservation conditions for organic remains [154]. Just as in Hornstaad–Hörnle IA, the material concerned (find no. Si10 538/127-1054, Fig 11) was taken from burnt layer 9 (2.9.2) which also most probably resulted from a catastrophic fire dated to the Middle Pfyn culture (1st half of the 37rd c. BCE)[155]. The given find number comprises five amorphous charred objects of irregular shapes and considerable sizes (the smallest 24 x 28 x 30 mm, the largest 68 x 53 x 39 mm). The largest chunk was documented via photogrammetry (see supplementary information) and then sampled for microscopical analysis. Another lump of processed cereals from the same layer, bearing a resemblance to a detached food crust and containing macroscopically visible chunks of cereals (find no. Si10 538/128-1030, Fig 12), was analysed as a reference.

**Zürich—Parkhaus Opéra.** This is the third late Neolithic lakeshore settlement included as a case study–also a site with waterlogged conditions and extraordinary preservation of organic finds. It has been studied in great detail in an interdisciplinary project [156–158]. Special emphasis was laid on the taphonomy [159], while the very rich botanical remains were and still are object of intense research [58, 160]. Settlement remains were attributable to seven very short-lived and dendrochronologically dated occupation phases between 3234 BC and 2727 BC [161]. Taphonomic studies revealed that, here again, the vast majority of the organic remains represented everyday refuse which can still be attributed to individual buildings. The object in question (find no. ZHOPE 12162.1A / AOV 85, Fig 13) was found in layer 14 dating to c. 3090 BC, located beneath house 401 [162]. Find no. ZHOPE 6949.1 was analysed as a reference. It was located near the southern wall of house 318, belonging to the older settlement layer 13, dated to 3176 BCE– 3153 BCE [163].

## Sample preparation and SEM analysis of the archaeological materials

Fragments of the brewing residues from Hierakonpolis were mounted and gold-sputtered prior to observation under high vacuum using a FEI Quanta 250 FEG SEM at the National Research Centre, Egypt. For the purpose of this study, a series of charred organic lumps found in one of the vats in the Tell el-Farkha brewing installations (feature 192/201) was subjected to additional SEM examination, involving multiple areas per amorphous charred object. The specimens were mounted on SEM stubs using carbon cement and subsequently platinum-palladium coated and examined using a JEOL-JSM-6480LV at magnifications of 40x to 1400x. SEM images of the archaeological finds from the sites of Haselbach, Hornstaad–Hörnle IA, from Sipplingen–Osthafen and from Zürich–Parkhaus Opéra were produced with the same devices and using the same preparation procedures as the experimentally charred barley malt.

## Measurements and statistical evaluation

Cell wall thickness was manually recorded using the software ImageJ [164] by measuring the double cell wall thicknesses (i. e. the shortest distance connecting two adjacent aleurone cell lumina) in SEM images (Fig 14). Only one distance per cell-cell border was recorded, thereby

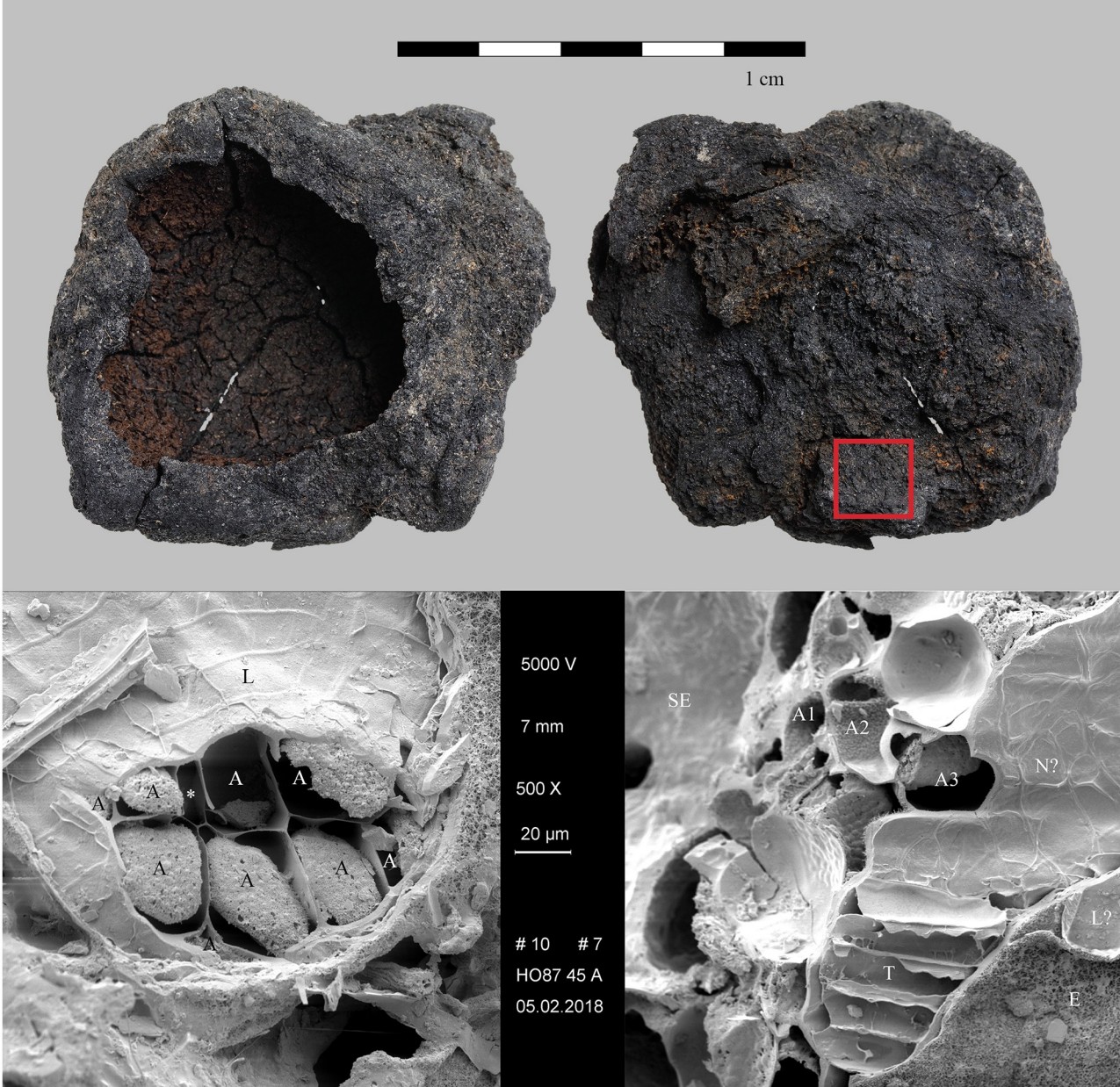

**Fig 10. The bowl-shaped charred cereal product ("*brei mit napfförmiger oberfläche*") from Hornstaad—Hörnle IA.** Find no. Ho 45/43-28. Top: light micrograph (red square: location of SEM subsample), bottom: SEM images. Left: patch of regularly arranged aleurone cells (A) with a conspicuous intercellular space (*) in between. L. . . longitudinal cells, right: fracture through the outer caryopsis layers, the multiple aleurone layers (A1 –A3) identify the material as cultivated barley (*Hordeum vulgare*) as do the thin-walled transverse cells (T). SE. . . starchy endosperm (fused remains), N? . . . probably nucellus tissue, L?. . . probably longitudinal cells, E. . . epidermis (abraded).. Images: ÖAW-ÖAI / N. Gail (light micrograph), A. G. Heiss (SEM). See also S1 Model.

measuring each cell wall only once. In cases where only the marginal (widened) parts of the cell walls were preserved due to abraded surfaces in the archaeological specimen, no measurement was carried out of the respective double cell walls in order to avoid biased measurements. The raw data of all measurements from the study are accessible in S1 and S2 Tables. The SEM images used to generate these values are accessible in the S1 and S2 Archives, respectively.

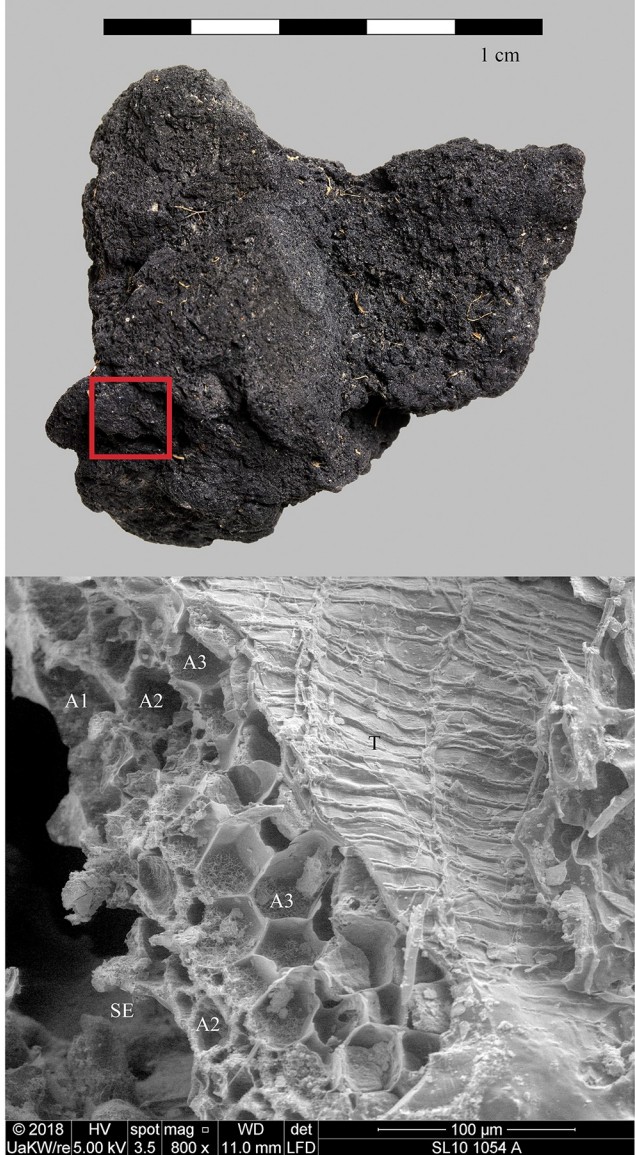

**Fig 11. Charred cereal product ("*feiner brei*") from Sipplingen—Osthafen.** Find no. Si10 538/127-1054. Top: light micrograph (red square: location of SEM subsample), bottom: SEM image, fracture through the outer caryopsis layers. The multiple aleurone layers (A1–A3) identify the material as cultivated barley (*Hordeum vulgare*). SE. . . starchy endosperm (fused remains), T. . . transverse cells. Images: ÖAW-ÖAI / N. Gail (light micrograph), die Angewandte / R. Erlach (SEM). See also S2 Model.

All statistical tests were performed using the software PAST 3 [165]. For descriptive statistics, see S3 Table. Normality tests (W, A, L, and JB, see S4 Table) indicated that the populations of most measurements were not normally distributed, suggesting the use of non-parametric tests for statistical evaluation. A Kruskal-Wallis test was therefore applied to all measurement series, which generated significant differences between groups (H ($\chi^2$): 1041; $H_c$ (tie corrected): 1041; p = 3.078$E^{-216}$). It was followed by Dunn's pairwise tests on raw p values with sequential Bonferroni significance (Table 1). Boxplots created with PAST 3 were processed for publication using Adobe Illustrator 6.

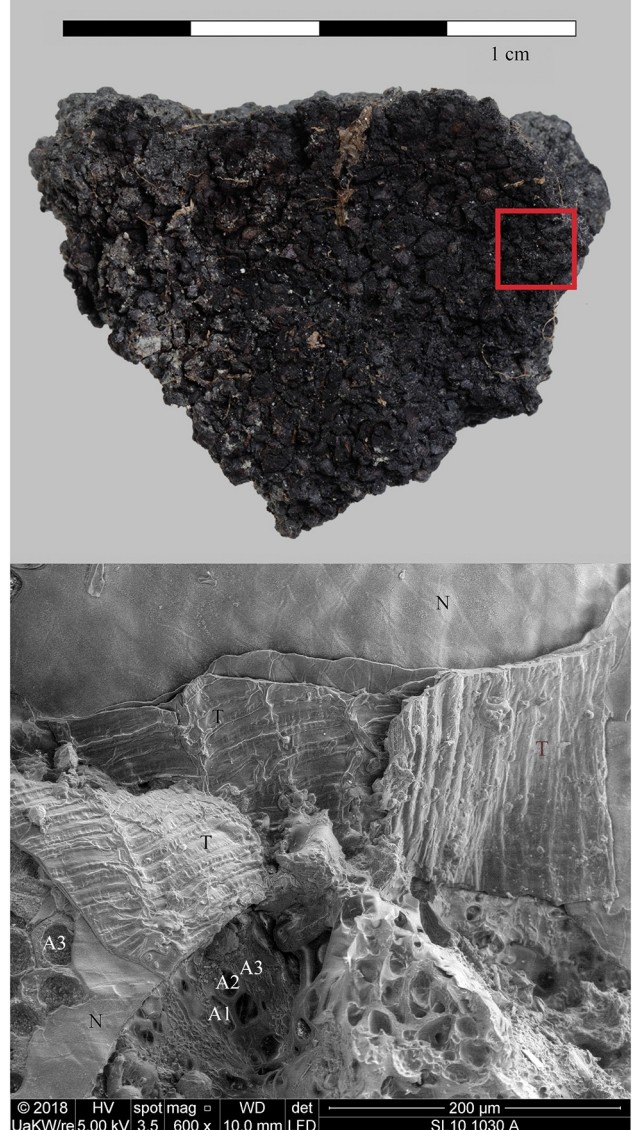

**Fig 12. Charred cereal product with large chunks of grains ("*grober getreidebrei*") from Sipplingen—Osthafen.**
Find no. Si10 538/128-1030. Top: light micrograph (red square: location of SEM subsample), bottom: SEM image,
fracture through the outer caryopsis layers. The multiple aleurone layers (A1–A3) identify the material as cultivated
barley (*Hordeum vulgare*). N. . . nucellus tissue, T. . . transverse cells. Images: ÖAW-ÖAI / N. Gail (light micrograph),
die Angewandte / R. Erlach (SEM).

## Results

### Degraded/thinned aleurone cell walls in the experimentally charred barley malt

In our experiments, we were able to demonstrate that the phenomenon of aleurone cell wall
degradation is clearly observable not only in fresh but also in charred material after five days of
previous sprouting. The double cell wall thickness in commercially available barley (*Hordeum
vulgare*) malt shows a progressive decrease (Fig 15, grey boxes), with significant shifts in mini-
mum, mean and maximum values from 0.76–(1.942)–4.2 µm after one day to 0.19 –(1.048)–

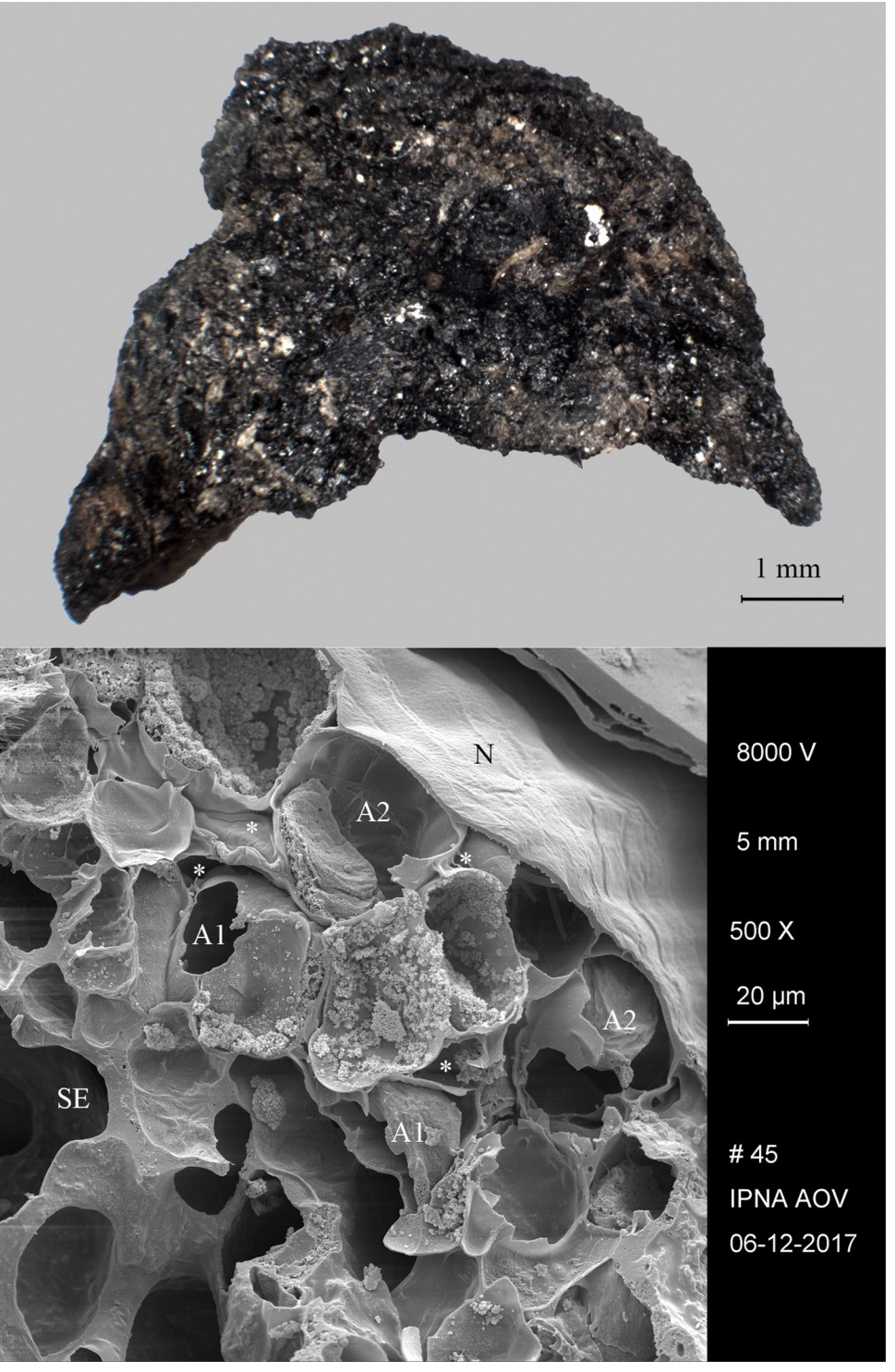

**Fig 13. Charred cereal product from Zürich—Parkhaus Opéra.** Find no. ZHOPE 12162.1A / AOV 85. Top: light micrograph, bottom: SEM image, fracture through the outer caryopsis layers. The multiple aleurone layers (A1–A2) identify the material as cultivated barley (*Hordeum vulgare*) and show conspicuous intercellular spaces (*). SE. . . starchy endosperm (fused remains), N. . . nucellus tissue. Images: UNIBAS-IPNA / F. Antolín.

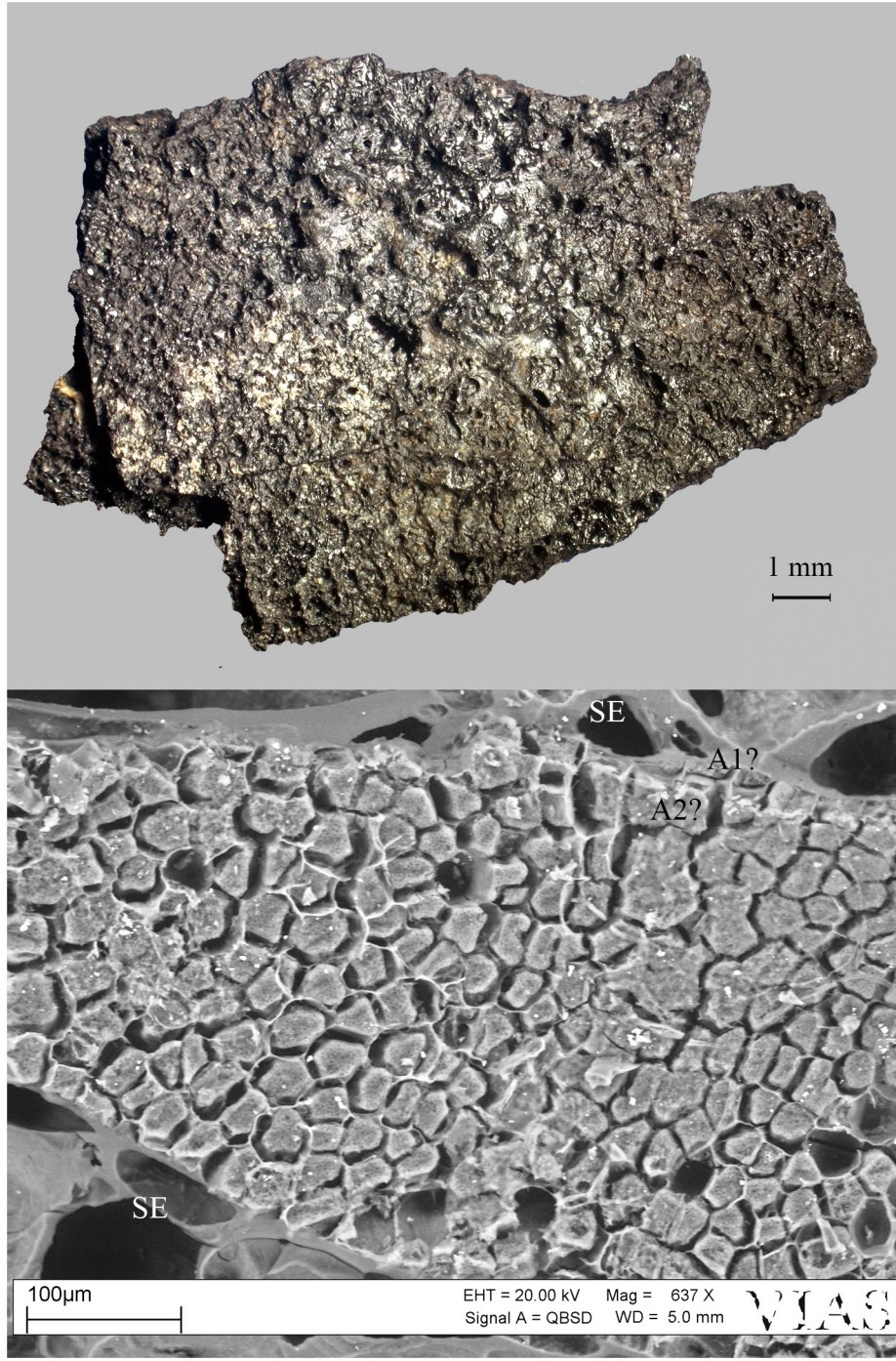

**Fig 14. Charred cereal product from Zürich—Parkhaus Opéra.** Find no. ZHOPE 6949.1. Top: light micrograph, bottom: SEM image, planar view of the aleurone layer. The presence of a single vs. multiple layers (A1? A2?) of aleurone is not clearly visible, therefore not allowing for precise identification of the material (cf. *Hordeum vulgare*). Images: ÖAW-ÖAI / A. G. Heiss (light micrograph), univie-VIAS / M. Mehofer & A. G. Heiss (SEM).

2.06 μm after five days of sprouting (S3 and S4 Tables). The statistical tests (Table 1) resulted in significant differences between the groups, with the exception of day 1 vs. day 2, and day 3 vs. day 4 of germination. According to the literature (see Introduction), much more drastic results are to be expected in grains which have sprouted for longer periods [113].

**Table 1. Pairwise differences between all measured groups (Dunn's test, raw p values, sequential Bonferroni significance).** p < 0.05 (= significant differences observed between groups) highlighted in pink.

| | exp. malt 1d | exp. malt 2d | exp. malt 3d | exp. malt 4d | exp. malt 5d | HK 11C | TeF 192/201 | Ho 45/43-28 | Si10 538/127-1054 | Si10 538/128-1030 | ZHOPE 6949.1 | ZHOPE 12162.1A |
|---|---|---|---|---|---|---|---|---|---|---|---|---|
| exp. malt 1d | | 0.073 | 1.03 E-06 | 3.33 E-07 | 1.54 E-31 | 2.76 E-53 | 3.97 E-51 | 2.21 E-58 | 1.86E-61 | 0.221 | 0.964 | 3.66 E-59 |
| exp. malt 2d | 0.073 | | 0.002 | 0.001 | 9.92 E-22 | 3.68 E-43 | 9.842 E-41 | 1.51 E-47 | 4.72E-49 | 0.009 | 0.134 | 8.94 E-49 |
| exp. malt 3d | 1.03 E-06 | 0.002 | | 0.837 | 9.97 E-11 | 2.02 E-30 | 7.70 E-28 | 5.58 E-34 | 5.32 E-34 | 5.92 E-07 | 7.03 E-05 | 1.35 E-35 |
| exp. malt 4d | 3.33 E-07 | 0.001 | 0.838 | | 3.68 E-10 | 1.05 E-29 | 4.05 E-27 | 3.23 E-33 | 3.63E-33 | 2.49 E-07 | 3.39 E-05 | 7.53 E-35 |
| exp. malt 5d | 1.54 E-31 | 9.92 E-22 | 9.97 E-11 | 3.68 E-10 | | 2.50 E-12 | 5.06 E-10 | 1.79 E-14 | 6.88E-13 | 2.07 E-23 | 5.37 E-20 | 2.57 E-16 |
| HK 11C | 2.76 E-53 | 3.68 E-43 | 2.02 E-30 | 1.05 E-29 | 2.50 E-12 | | 0.335 | 0.662 | 0.495 | 2.62 E-43 | 1.03 E-39 | 0.228 |
| TeF 192/201 | 3.97 E-51 | 9.84 E-41 | 7.70 E-28 | 4.05 E-27 | 5.06 E-10 | 0.335 | | 0.153 | 0.727 | 9.91 E-41 | 3.90 E-37 | 0.029 |
| Ho 45/43-28 | 2.21 E-58 | 1.51 E-47 | 5.58 E-34 | 3.23 E-33 | 1.79 E-14 | 0.662 | 0.153 | | 0.242 | 9.47 E-47 | 4.06 E-43 | 0.427 |
| Si10 538/127-1054 | 1.86 E-61 | 4.72 E-49 | 5.32 E-34 | 3.63 E-33 | 6.88 E-13 | 0.495 | 0.727 | 0.242 | | 2.09 E-46 | 9.63 E-43 | 0.048 |
| Si10 538/128-1030 | 0.221 | 0.009 | 5.92 E-07 | 2.49 E-07 | 2.07 E-23 | 2.62 E-43 | 9.91 E-41 | 9.47 E-47 | 2.09E-46 | | 0.304 | 2.06 E-48 |
| ZHOPE 6949.1 | 0.964 | 0.134 | 7.03 E-05 | 3.39 E-05 | 5.37 E-20 | 1.03 E-39 | 3.90 E-37 | 4.06 E-43 | 9.63E-43 | 0.303 | | 8.57 E-45 |
| ZHOPE 12162.1A | 3.66 E-59 | 8.94 E-49 | 1.35 E-35 | 7.53 E-35 | 2.57 E-16 | 0.227 | 0.029 | 0.427 | 0.049 | 2.06 E-48 | 8.57 E-45 | |

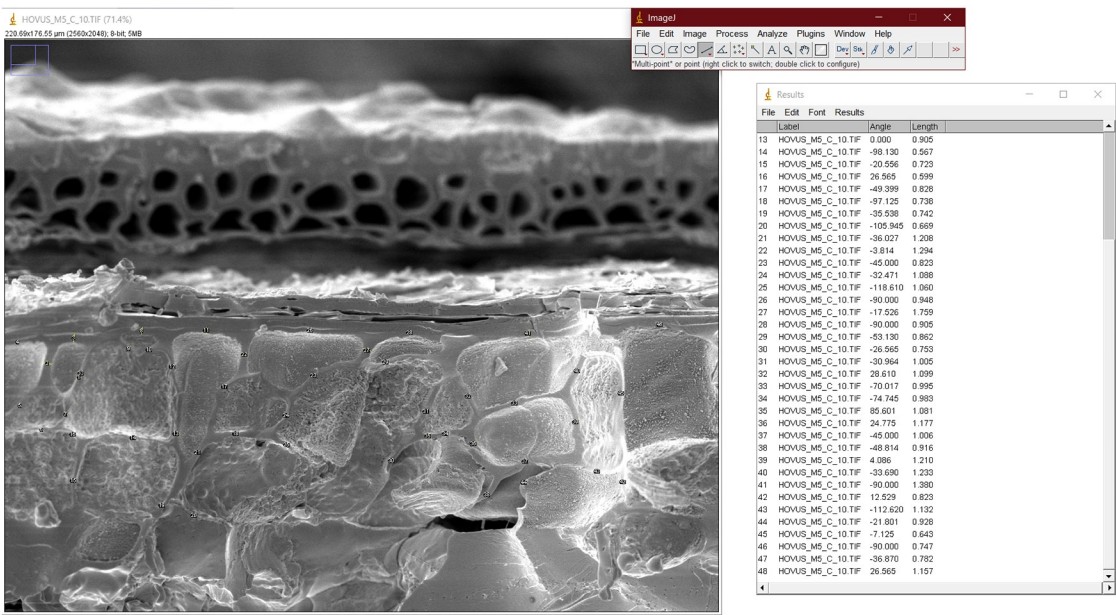

**Fig 15. Example of data acquisition.** Measuring aleurone cell walls (double cell wall thickness) in experimentally charred barley (*Hordeum vulgare*) malt. Image: ÖAW-ÖAI / Andreas G. Heiss.

## Degraded/thinned aleurone cell walls in the archaeological materials

The effect observed in the experimentally charred modern malt was not only confirmed for the materials from the archaeological case studies but it was even much more pronounced (Fig 15). While the means and medians of double cell wall thickness in modern barley malt amounted to c. 1.3–1 μm in the period of 3 till 5 days of sprouting, they never exceeded 0.7 μm in the archaeological specimens. Factors possibly influencing cell wall thickness other than sprouting (see Discussion section) have successfully been ruled out for all sites. None of the specimens displayed marked infestation with mycelial fungi, perforated cell walls, an aleurone entirely depleted of its contents, or cell wall thinning observable in the outer layers of the grain.

We therefore conclude that materials from all five archaeological sites reveal clear evidence of sprouted grains which were later ground and transformed into a food preparation. The markedly thin-walled emmer (*Triticum dicoccum*) aleurone patches from the known brewing installations at Hierakonpolis (Fig 8) and Tell el-Farkha (Fig 9) are a clear confirmation of our experimentally supported suggestions for detecting malt in charred archaeological cereal products, possibly indicating sprouting periods exceeding five days. The barley (*Hordeum vulgare*) aleurone tissues found at Hornstaad–Hörnle IA (Fig 10), Sipplingen–Osthafen (Si10 538/127-1054, Fig 11), and Zürich–Parkhaus Opéra (ZHOPE 12162.1A, Fig 13) also clearly confirm the presence of malted cereal grains as the basis of the analysed cereal products and sprouting seems to have taken place to a similar extent as in the Egyptian finds. In contrast, the reference samples of food crusts from Sipplingen–Osthafen (Si10 538/128-1030, Fig 12) and from Zürich–Parkhaus Opéra (ZHOPE 6949.1, Fig 16) do not display any marked traces of aleurone thinning

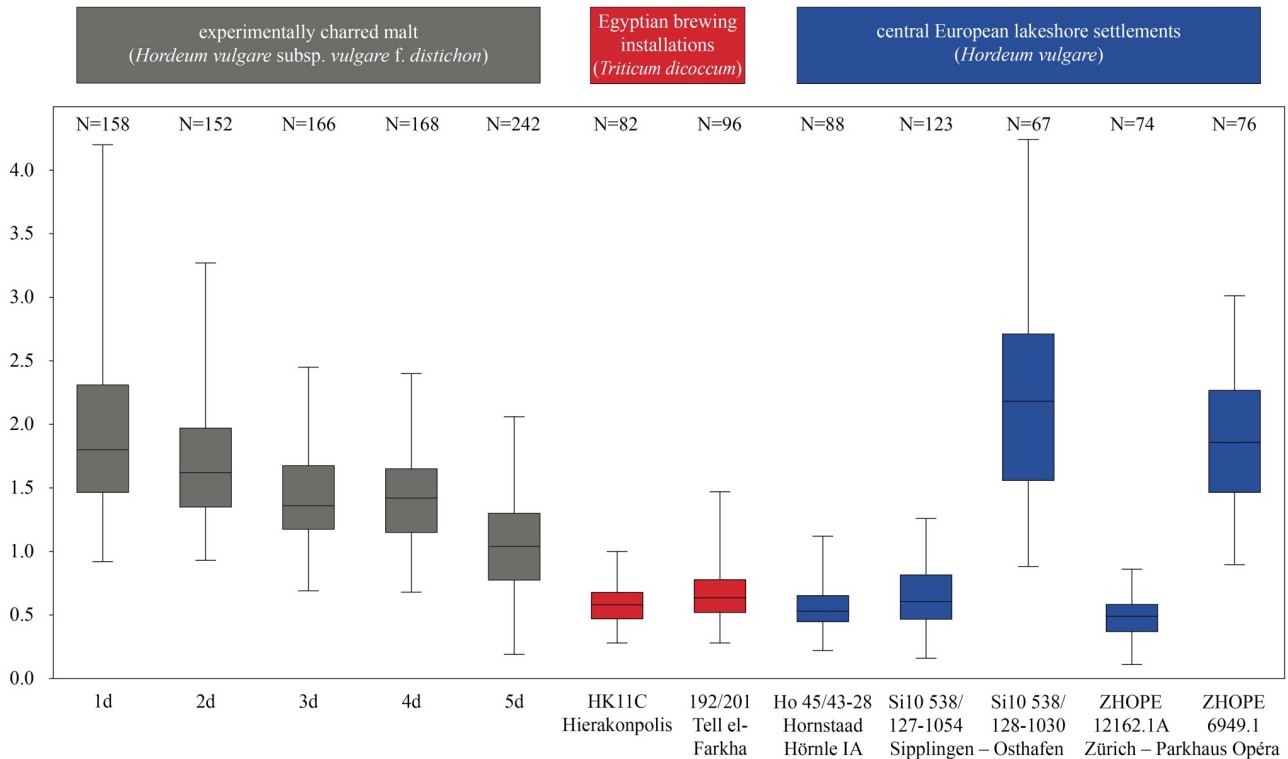

**Fig 16. Aleurone double cell wall thicknesses in μm, observed in experimentally charred barley malt, compared to the archaeological finds of charred cereal preparations included in this study.** For descriptive statistics of the raw data see S3 Table, for normality tests see S4 Table, for post hoc tests see Table 1. Figure: ÖAW-ÖAI / A. G. Heiss.

and thus malting. Furthermore, statistical analysis shows that these two archaeological reference samples have aleurone wall thicknesses comparable to a virtually unsprouted state, i. e. the 1d and 2d sprouted experimental barley malt (Table 1, Fig 15).

## Discussion

### Possible pitfalls in the interpretation of thin-walled aleurone tissue

Apart from sprouting leading to cell wall degradation, also other processes need to be taken into account when interpreting thinned aleurone cell walls in archaeological finds of processed foodstuffs.

1. **Fermentation by mycelial fungi**. The lacking or very low cellulase activity in yeasts as well as in LAB and AAB as mentioned in the Introduction allows ruling them out as significant factors for cell wall hydrolysis. Technical applications relying on fast cell wall degradation therefore usually involve cellulases extracted from saprobiontic fungi [166] such as *Trichoderma reesei* [167] or *Acremonium thermophilum* [168]. Likewise, kōji fungi such as *Aspergillus flavus* var. *oryzae* display notable cellulase and xylanase activity [169].
   Rather than a colonisation with yeasts, AAB, and LAB, it is rather an ancient foodstuff's infestation with mycelial fungi prior to charring which must be considered as an additional possible cause for conspicuously degraded aleurone cell walls in a charred cereal product. At the same time, however, such material would also display a) massive occurrence of fungal hyphae, b) perforated cell walls, and c) aleurone cells depleted of their contents. Finds in question should therefore always be checked against the presence of these three indicators of fungal decay. However, none of the aforementioned was observed in the materials included in the current study.

2. **Endogenous enzymes**. As mentioned in the Introduction, it is important to stress that the largest part of enzymes responsible for cell wall breakdown are not constantly present in the aleurone layer but are synthetized *de novo* during germination [99]. The unsprouted grain's activity of endogenous xylanases and 1,4-β-glucosidases [115] is reportedly very low [170] and would require several weeks of fermentation to have a noticeable effect–a reason why modern bread production uses artificially added cellulases as components of dough conditioners if timely cell wall breakdown is desired [171–173].
   In the case of traditional stable sourdoughs, which can be kept metabolically active for decades [174]–allegedly even for centuries [175]–by the repeated addition of substrate and water, endogenous hydrolytic enzymes certainly have enough time to digest most of the cell wall material. During the same time, the sourdough's microbiome with its enzymatic activity could bring about digestion of the available nutrients from the former cereal grain material [176–178] in these aged sourdough systems. This possibility of aleurone thinning reflecting very mature sourdough in the archaeobotanical finds needs to be systematically investigated with experimental material in the future. Notwithstanding, the aleurone cells in a cereal-based ACO should always be checked against depleted cell contents in order to exclude such a scenario (for further discussion, see below). In the current study, however, aleurone cells with intact cell content were still observed.

3. **Thermal degradation during charring**. At temperatures significantly exceeding 300 ˚C, thermal degradation of hemicelluloses [145] and of cellulose [146] causes massive loss in substance–and therefore cell wall thickness–typically amounting up to 30%. This effect would however be observable in all the grain's tissues to the same extent. The materials in question therefore need to be checked against conspicuously "thinned-out" cell walls in

other parts of the cereal bran (e. g., transverse cells). If no tissues for comparison are available in an archaeological cereal product, extra care needs to be taken before reaching any interpretation of sprouting/malting.

## Validity of the observations in the analysed materials

Our results clearly show that even small patches of aleurone tissue preserved in amorphous charred food remains can be used to document sprouted and probably malted cereal grains, especially when the grains contained therein have been thoroughly crushed or ground and soaked in liquid prior to charring. The underlying phenomenon of endosperm and aleurone cell wall degradation increases with the duration of sprouting and it is more pronounced at the embryo end of the grain, while it may not develop at all at the distal end (see Introduction). We tried to overcome this by observing tissues in the middle section of the experimentally charred barley malt, knowing that the effects would diverge strongly towards the grains' ends. As a straightforward consequence of our observations, it has to be stated that, while the presence of significantly thinned cell walls and intercellular spaces in the aleurone layer are indicative of several days of germination, their absence allows no conclusions whatsoever on the presence or absence of malt in a charred ground cereal preparation.

## Perspectives on a quantification of the effect: How thin is "thin"?

Our experiments were conducted only on barley, knowing that aleurone cell walls in wheat species can be up to two thirds more massive (see Introduction). The reference material which was used had only been sprouted for five days, while more notable effects are to be expected after longer sprouting periods. It must therefore be stressed that our results from experimentally charred malt are most probably biased towards an underestimation of the potential maximum extent of cell wall degradation in a sprouting grain.

When looking at the data from the archaeological materials from the five case studies, they indeed seem to indicate a sprouting period exceeding the five days of the modern reference material. However, as the former positions of the observed aleurone tissues in the grains are entirely unknown in the charred ground cereal preparations, the perspective of estimating or quantifying the sprouting time for archaeological malt finds seems unrealistic. At the very least, a large series of experimental malting and consecutive charring under varying regimes would be required in order to approximate such a quantitative approach.

Until then, we suggest that the discrimination between sprouted and non-sprouted grains should only be applied in a qualitative sense. As a robust microstructural indicator of sprouting/malting, we suggest a threshold value not exceeding a mean/median of 1 μm for double cell wall thickness, together with an upper quartile not exceeding 1.5 μm.

## "Egyptian ale" and "lakeshore lager"? Take them with a pinch of. . . malt!

As for the functional interpretation of the materials included as case studies, the situation is strongly divergent between the Egyptian and the central European finds. The thinned aleurone cell walls in the Hierakonpolis and Tell el-Farkha material have to be considered to the ample archaeological evidence for beer production at both sites. As a consequence, the malt-based amorphous charred objects (ACO) which were recovered from the inside of the heatable vats of the brewing installations can safely be classified as brewing remains, most probably from mashing (step no. 6 as illustrated in Fig 1). Whether any alcoholic fermentation had taken place prior to charring is, however, beyond our methodological grasp.

In contrast, the same aleurone thinning observed in the finds from the three central European Neolithic lakeshore settlements is not directly related to clear evidence–be it contextual or artefactual–for brewing activities. So, how much hard evidence for brewing do we really have? Beer-making from sprouted grains seems to be nearly ubiquitous in the ethnographic, historical, and archaeological record. To a large extent, this popularity is certainly attributable to the pleasant effects of alcohol [3, 37, 41, 47, 179], but also to beer's hygienic benefits. As its production often involves cooking and thus sterilisation, beer has been a much less harmful drink than water until the modern period. The contamination with intestinal parasites of the waters around the lakeshore settlements has already been impressively demonstrated for several central European Neolithic sites [180, 181] and most recently for Parkhaus–Opéra [182]. The inhabitants of these settlements definitely had good reason to produce and consume beer. The find situations of the charred cereal preparations basically support such ideas. While charred flat breads and verified dough remains are rare in relation to the total quantity of plant remains recovered from the respective sites, most of the ACO found–including the ones analysed in the current study–come from the inside of cooking vessels. Cereal preparations not involving any heating process, such as the long-term storage of sourdough, are therefore less probable as a source of these materials, in addition to the structural features already discussed above.

A look at the initially proposed *chaîne opératoire* of "core" processes in brewing (for the numbers, see Fig 1) shows that the materials in question had undergone 1) soaking and 2) sprouting, indicated by the presence of degraded aleurone cell walls. The sprouted grains, likely after 3) drying or roasting, had then been 5) crushed or ground, the resulting malt meal or malt flour being 6) soaked again at one point, and mixed into a more or less homogeneous mass. The malt-containing remains from Sipplingen–Osthafen and Zürich–Parkhaus Opéra bear no clear hints as to the consistency of the resultant mixture, opening a wide spectrum of possible malty outcomes–liquid, solid, and everything in between. The cup-shaped ACO (in fact, a bread-like object [58]) from Hornstaad-Hörnle IA, however, displays features typically found in charred remains of liquids, such as a cracked surface and conspicuous size sorting of the particles [57, 183] (Fig 17)–structural traits which it also shares with the brewing residues from Hierakonpolis and Tell el-Farkha. Furthermore, experiments support scenarios leading to the formation of the Hornstaad object's unusual shape from an initially liquid food preparation [183] (Fig 18). We can be certain that the charred find from Hornstaad–Hörnle IA derives from some malt-containing drink.

## Beyond the horizon: Food complexity's uncharted territory...

If we widen the perspective beyond beer, already a few quick glances into the (ethno-)historical treasure chest reveal a variety of malt-based foods beyond beer. The Austrian 17[th] century author E. M. R. von Liechtenstein, for example, mentions barley malt as a tonic for pregnant women [184]. The production of malt bread [185] and malt syrup [186] is documented for late 18[th] century Germany in Krünitz' *Oeconomische Encyklopädie*, with this source recommending [185, 186] malt-based drinks [187] as tonics [187, 188]. In the 19[th] century, A. Maurizio's famous book "*Die Getreide-Nahrung im Wandel der Zeiten*" unfortunately remains inconclusive on the topic of malt, as the author refers to both sprouted and unsprouted grain as *Malz*, (e.g, p. 46) [189]. The French author C. L. Husson promotes malt as an ingredient for weaning foods [190], a use which is also of contemporary relevance [191, 192]. It is these first impressions that strongly suggest the need for the development of an in-depth study on malt-based foodstuffs from Antiquity until today in order to provide a basis of discussion for the interpretation of archaeological remains of malt-containing cereal preparations.

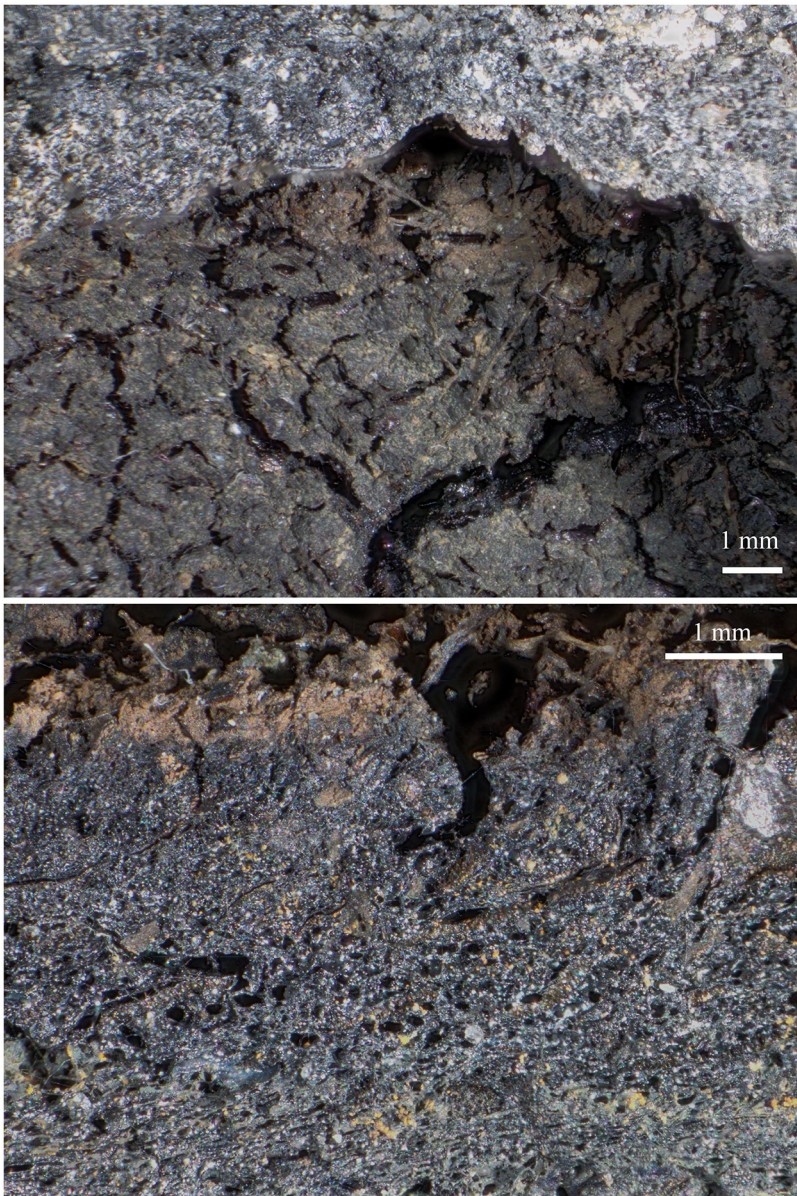

**Fig 17. Surface features (top) and cross section (bottom) of the bowl-shaped charred cereal product from Hornstaad—Hörnle IA.** The distinct cracked surface (*craquelé* [57]) pattern as well as the particle size sorting in the material are interpreted as resulting from a liquid state of the material prior to charring. Image: ÖAW-ÖAI / A. G. Heiss.

Furthermore, it is a characteristic of culinary practices that basically every end product of an operational sequence can become the initial product of another one, adding much complexity to the possible interpretations of archaeological food remains [57]. Taking bread as an example, this end product can be dried and stored and eventually become an ingredient of soup prior to consumption [193]. Bread can be ground and mixed with flour for the production of new bread [194, 195]. Bread can serve as a starter in beer making [4, 35, 42]. Spent

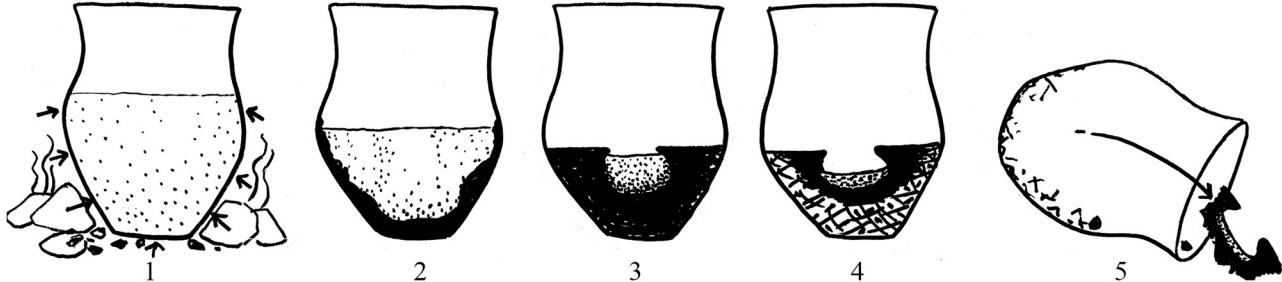

**Fig 18. Model for the formation of the bowl-shaped bread-like object from Hornstaad-Hörnle.** 1) cooking pot in regular use, surrounded by heated stones. During either a cooking accident, or in the catastrophic fire, 2) the heat causes the pot's liquid content to boil, to dry out and to char progressively from the outside inwards until 3) only the final remains of the liquid are left in the centre. 4) All liquid is now dried out, and the outermost parts of the mass possibly even begin to burn to ashes (hatched areas). 5) The bowl-shaped innermost part of the charred mass breaks loose as soon as the pot falls over or is destroyed, or intentionally emptied and cleaned. Image: RPS-BWL / H. Schlichtherle. Modifications: ÖAW-ÖAI / A. G. Heiss.

grain, in turn, can be used as raw material for bread production [56, 171]. Considering this, the possibilities of how to further process and consume germinated grain increase even more.

We can certainly speculate that either only the Hornstaad food preparation or even all of the malt-containing finds from the lakeshore settlements could indeed have derived from beer. Unfortunately, due to the lacking diagnostic tools for identifying the actual process of fermentation, any such claim is to be taken with a huge pinch of salt and remain what it is–speculation.

We can, however, clearly state that the processing of malted grain has been practiced as part of various food production sequences at the central European Neolithic lakeshore settlements in the 4th millennium BCE.

## Conclusions

In archaeology, the statement "We have evidence of beer!" has far-reaching consequences for the materials concerned, for the respective contexts and sites and even beyond [7, 8]. While with great certainty we can claim that the samples from the two Egyptian brewing installations indeed directly derive from the brewing process, the lack of equally unambiguous context renders similarly clear conclusions somewhat dangerous for the materials from the central European lakeshore settlements. The sample from Hornstaad–Hörnle very likely represents the remains of some malt-containing liquid, the kind of which must remain uncertain. The malt-containing amorphous crusts from Sipplingen–Osthafen and Zürich–Parkhaus Opéra derive from either liquid or solid foodstuffs, one ingredient of which was malt. The find situations inside cooking vessels, however, tend to substantiate the hypothesis of liquid foods.

We are confident that the observations presented here are a significant leap forward in the research of food history in general and brewing history in particular. They provide a new diagnostic feature for the detection of processed sprouted (malted) grains in amorphous charred objects and food crusts even if no intact grains are left, even in sites where no further unambiguous archaeological, epigraphic, iconographic, or other evidence is preserved.

We propose the observed feature of degraded/thinned aleurone cell walls as a novel diagnostic marker in the research of ancient cereal processing and we encourage colleagues to re-evaluate extant SEM images of aleurone tissue from archaeological contexts in search of this practice, in order to help broaden our knowledge on past malting and possible brewing.

## Supporting information

**S1 Model. Photogrammetric 3D model of the bowl-shaped cereal product from Hornstaad —Hörnle IA.** Wavefront OBJ file [196]. Also accessible at https://sketchfab.com/models/b96ab3c0db0d41978e0adf7f92de54c1.
(OBJ)

**S2 Model. Photogrammetric 3D model of the largest amorphous fragment of the cereal product from Sipplingen—Osthafen.** Wavefront OBJ file [197]. Also accessible at https://sketchfab.com/models/db37ab7cae4c46288b5912c2ae7e49c6.
(OBJ)

**S1 Archive. SEM images used for the generation of the values in S1 Table.**
(ZIP)

**S2 Archive. SEM images used for the generation of the values in S2 Table.**
(ZIP)

**S1 Table. Aleurone cell wall measurements of the experimentally charred barley malt, raw data.**
(XLSX)

**S2 Table. Aleurone cell wall measurements of the archaeological finds of charred cereal preparations, raw data.**
(XLSX)

**S3 Table. Descriptive statistics for all measured items.**
(XLSX)

**S4 Table. Normality tests for all measured items.**
(XLSX)

## Acknowledgments

The authors are greatly indebted to the following participants (in alphabetic order) of the international workshop "Ancient beer: multidisciplinary approaches for its identification in the archaeological record" held at the University of Hohenheim in February 2019 for their immensely valuable and fruitful discussions: Bettina Arnold and Joshua I. Driscoll (both University of Wisconsin-Milwaukee), Ralf Kölling-Paternoga (University of Hohenheim), Maxime Rageot (University of Munich), Eva Rosenstock (Freie Universität Berlin), and Delwen Samuel (Royal College of Occupational Therapists). We are grateful to Véronique Zech-Matterne (MNHN Paris, UMR 7209 du CNRS) and to Mustafa Bayram (Gaziantep University) for their valuable suggestions. Our warmest thanks go to Erika Rücker and Anne Heller (both University of Hohenheim), to Rudolf Erlach (University of Applied Arts Vienna), to Mathias Mehofer (University of Vienna), and to Elshafaey A. E Attia (Helwan University, Cairo) for their help in producing SEM images, and to Niki Gail and Christian Kurtze (both ÖAW-ÖAI) for producing the photogrammetric 3D models. We are thankful to Jörg Helbing (Unterberger Automation, Nüziders) for kindly supporting us with his CLSM images of malted cereals. We thank Renate Ebersbach (State Office for Cultural Heritage Baden-Württemberg) for kindly inviting the first author to the *Hauskolloquium* at Hemmenhofen-Gaienhofen in February 2019. The authors are grateful to the Heinrich Durst Malzfabriken GmbH & Co. KG (Bruchsal-Heidelsheim) for kindly providing sprouted barley for our research. We are also grateful to the academic editor, Ceren Kabukcu, the reviewers Mark Nesbitt and the anonymous one, who

provided useful comments and suggestions to improve the manuscript. We are grateful for English language editing to Bisserka Gaydarska, John Chapman and John Gorczyk. We thank the late David Bowie for his œuvre, and in particular for giving us Major Tom.

## Author Contributions

**Conceptualization:** Andreas G. Heiss, Marian Berihuete Azorín.

**Data curation:** Andreas G. Heiss.

**Formal analysis:** Andreas G. Heiss.

**Funding acquisition:** Andreas G. Heiss, Ferran Antolín, Hans-Peter Stika, Masahiro Baba, Niels Bleicher, Krzysztof M. Ciałowicz, Marek Chłodnicki, Irenäus Matuschik, Helmut Schlichtherle, Soultana Maria Valamoti.

**Investigation:** Andreas G. Heiss, Marian Berihuete Azorín, Ferran Antolín, Lucy Kubiak-Martens, Elena Marinova, Hans-Peter Stika.

**Methodology:** Andreas G. Heiss, Marian Berihuete Azorín, Hans-Peter Stika.

**Project administration:** Andreas G. Heiss, Ferran Antolín, Hans-Peter Stika, Soultana Maria Valamoti.

**Resources:** Andreas G. Heiss, Marian Berihuete Azorín, Ferran Antolín, Lucy Kubiak-Martens, Elena Marinova, Hermann Kretschmer, Hans-Peter Stika, Niels Bleicher, Krzysztof M. Ciałowicz, Marek Chłodnicki.

**Supervision:** Andreas G. Heiss, Soultana Maria Valamoti.

**Validation:** Andreas G. Heiss, Marian Berihuete Azorín, Lucy Kubiak-Martens, Elena Marinova, Elke K. Arendt, Costas G. Biliaderis, Hermann Kretschmer, Athina Lazaridou, Hans-Peter Stika, Martin Zarnkow, Masahiro Baba, Niels Bleicher, Krzysztof M. Ciałowicz, Marek Chłodnicki, Irenäus Matuschik, Helmut Schlichtherle, Soultana Maria Valamoti.

**Visualization:** Andreas G. Heiss, Marian Berihuete Azorín, Elena Marinova, Elke K. Arendt, Martin Zarnkow.

**Writing – original draft:** Andreas G. Heiss.

**Writing – review & editing:** Andreas G. Heiss, Marian Berihuete Azorín, Ferran Antolín, Lucy Kubiak-Martens, Elena Marinova, Hans-Peter Stika, Martin Zarnkow, Niels Bleicher, Soultana Maria Valamoti.

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
