## [Decision Letter · Decision Letter 0]

30 Oct 2019

PONE-D-19-25202

Mashes to Mashes, Crust to Crust. Presenting a novel and comprehensive microstructural marker for malting in the archaeological record, integrating experimental data with 4th millennium BCE archaeobotanical evidence from Egypt and central Europe

PLOS ONE

Dear Dr Heiss,

Thank you for submitting your manuscript to PLOS ONE. After careful consideration, we feel that it has merit but does not fully meet PLOS ONE’s publication criteria as it currently stands. Therefore, we invite you to submit a revised version of the manuscript that addresses the points raised during the review process.

This manuscript is of high scientific quality, the evidence provided is detailed and certainly merits publication in PLOS One. However, as pointed out by both reviewers, the manuscript and the conclusions within would benefit from greater detail on the identification of mashing, sprouting, malting and fermentation using the criteria outlined. To this end, Reviewer 2 has made specific suggestions for how the results and conclusions drawn in the manuscript can be strengthened. Following on from these comments, I would suggest to include some more measurements on un-malted grains from the archaeological sites included in the study (if these are available for measurements), to demonstrate better the contrast between aleurone thickness of malted vs. un-malted (and/or likely malted) grains. Furthermore, as suggested by both reviewers, please revise the sections relating to the identification of fermentation- as malting (or the identification of grain sprouting) is not always done for the purposes of fermentation. A more detailed discussion of the possible pathways of aleurone thinning (e.g., malt drinks, various levels of fermentation, food preparation involving sprouted grains) would certainly direct and enrich future work on archaeobotanical research relating to cereal food and beverage preparations in archaeology.

Both reviewers and myself are convinced that this has the potential to be a landmark publication in archaeobotany and any improvements suggested are likely to strengthen the scientific rigour and increase the applicability of the methodologies developed.

We would appreciate receiving your revised manuscript by December 15, 2019. To enhance the reproducibility of your results, we recommend that if applicable you deposit your laboratory protocols in protocols.io, where a protocol can be assigned its own identifier (DOI) such that it can be cited independently in the future. For instructions see: http://journals.plos.org/plosone/s/submission-guidelines#loc-laboratory-protocols

We look forward to receiving your revised manuscript.

Kind regards,

Ceren Kabukcu, PhD

Academic Editor

PLOS ONE

Journal Requirements:

2. In your manuscript, please provide additional information regarding the specimens used in your study. Ensure that you have reported specimen numbers and complete repository information, including museum name and geographic location.

For more information on PLOS ONE's requirements for paleontology and archaeology research, see https://journals.plos.org/plosone/s/submission-guidelines#loc-paleontology-and-archaeology-research.

"The authors have declared that no competing interests exist. ".

We note that one or more of the authors are employed by a commercial company: 'BIAX Consult,' and 'Braxar GmbH'.

Reviewers' comments:

Reviewer's Responses to Questions

**Comments to the Author**

1. Is the manuscript technically sound, and do the data support the conclusions?

Reviewer #1: Partly

Reviewer #2: Partly

2. Has the statistical analysis been performed appropriately and rigorously? 

Reviewer #1: Yes

Reviewer #2: I Don't Know

3. Have the authors made all data underlying the findings in their manuscript fully available?

Reviewer #1: Yes

Reviewer #2: Yes

4. Is the manuscript presented in an intelligible fashion and written in standard English?

Reviewer #1: Yes

Reviewer #2: No

5. Review Comments to the Author

Reviewer #1: This is an interesting manuscript that proposes the hypothesis that thinned aleurone layers of cereal grains are a marker for germination of cereals by design (malting) for making beer. The subject is certainly of interest in the cultural evolution of humans.

Data to support this hypothesis have been assembled from charred plant remains obtained from five archaeological sites in Egypt and Europe, and from model experiments. Contextual evidence of beer making was presented for the two sites in Egypt. The data collection and analysis seem thorough, the manuscript is clearly presented and well written, and it tells a cohesive story. Measurements of the width of aleurone cell layers in charred, sprouted barley grains (implicitly obtained under malting conditions) in the model experiments demonstrated a decrease in thickness over time, consistent with processes that occur during malting. The contextual evidence from the sites in Egypt, and the importance of beer in the pre-historic culture of this region, adds some weight to the proposal that the Egyptian samples provide evidence for malting and beer making. However in my opinion, the evidence from the archaeological samples from Europe is too sketchy to be indicative of beer brewing, and the evidence as a whole is insufficient to support a general claim of finding a marker that “can be considered in the research for ancient beer in any archaeological site where charred preservation occurs”.

Sprouted grains (most likely affected by microbial activity) have been used as foods in many cultures for as long as grains have been cultivated. A statement that malting was associated with food processing might seem more reasonable than linking it strongly to beer production, although that may have been an outcome. A greater concern I have relates to ancient cereal grains being much smaller than those from modern varieties (e.g., Ferrio et al. 2006. J Cer Sci44, 323-332; Helbelstrup 2017. Plant Sci 256, 1-4). The measurements taken from all of the archaeological samples, showed aleurone layers of similar thickness, but significantly smaller than those from the modern grains. This observation could be explained as structures coming from small grains rather than being indicative of germination. Grains are naturally associated with many active microbes, which as pointed out, could have caused the morphological changes. Hence, I am not persuaded by the statement of “thinned (degraded) aleurone cell walls as a general new marker for recognizing malting in cereal products from archaeological contexts”.

Overall, I find the manuscript could make a useful contribution to this interesting area of prehistoric human activity if the discussion, conclusions and abstract are recast and softened to be more balanced to alternative interpretations. The title is cumbersome and should be shortened … at the least, I suggest omitting the words “and comprehensive” and the words after “record”.

I also recommend the following minor editorial corrections.

1. The statement that the main protein content in cereal grains is in the aleurone layer (p6, line 185) is incorrect; aleurone cells contain protein bodies, but the majority of cereal grain proteins are storage proteins in the endosperm.

2. The term “heterocelluloses” is inaccurate and not meaningful; it should be replaced by heteropolysaccharides.

Reviewer #2: This paper presents an important advance in the detection of malting in ancient samples, moving beyond starch characteristics that usually disappear during charring. The aleurone layer often survives charring and is recognisable under SEM, meaning that the proposed criterion can be widely applied to charred and uncharred residues. The ability to detect malting in a wide range of archaeological contexts has the potential, as demonstrated through the case studies, to significantly advance our understanding of the dietary and social roles of food in ancient societies.

The paper is well-grounded in the literature of brewing science and extensive experimentation. Data is fully presented in the supplementary files and technical aspects of methodology are fully explained. Alternative explanations are fully considered. Although I make some suggestions on methodology below, I would expect the requested additional information to confirm the results of this study.

My suggestions for major changes are as follows:

Control samples:

- For the experimental work, I could not see the figures for aleurone thickness for the Heinrich Durst Malzfabriken barley grains *before* malting, either in table 1 or elsewhere – did I miss them? In the case of wheat, where there are control (unmalted) grains and malted grains (line 274), I could not find the aleurone thickness reported in the paper. For both wheat and barley it’s hard to asses the evidence for the effects of malting without seeing the measurements for malted vs unmalted grains.

- For the ancient residues, it should be standard practice to compare residues from ‘likely’ contexts with those from ‘unlikely’ contexts. For example, when evidence of food only occurs in association with food-related objects and is absent from tested non-food related objects or soil matrix, it strengthens the case for the age and identification of the food residue. In this instance the control sample could be grains from the sites or periods concerned that are intact and do not show signs of germination. Would it be possible to present some data on aleurone thickness from intact charred grains from the European sites? And for the Egyptian sites (material harder to access) at least from uncharred grains of a similar date in accessible collections. As with the experimental samples, comparing measurements for likely-malted vs unlikely-malted would strengthen the authors’ case.

Beer vs malted foods

135, Fig. 1 and various points in the manuscript. I think you could strengthen the distinction between mashing and fermenting (probably should be distinct steps in Fig 1). *It is only the last step that produces alcohol and is evidence for beer.* The criterion of the thinning of the aleurone layer is, if I understand correctly, evidence of saccharification, not of fermentation?

It might be worth adding a heading to explore the circumstances in which evidence of malting and grinding can be interpreted as evidence of beer. Of course human nature and Occam’s razor suggests this is often the case, but at the same time there is a risk inherent in projecting our current interest in beer back into the past. For example, in the paragraph starting 503, for example, one residue has a liquid appearance, leading to all being interpreted as beer. But couldn’t this be a malt drink? And the non-liquid food, malt foods? There is a strong tradition in several parts of the world of malted foods or beverages, notably in connection with weaning, and with millets in Africa. It would be worth adding a paragraph to explore this – there is plenty of literature on malted foods. What does the record look like for traditional food cultures in Europe? If malted foods and drinks are absent it strengthens the case for beer. Adam Maurizio might be good for this.

Another example is at 523 “As beer brewing based on sprouted grains is ubiquitous in the ethnographic, historical, and archaeological record, probability is very high that these remains indeed represent Neolithic beer mash.” is another example of projecting the ubiquity of beer today into the archaeological record, creating a self-fulfilling cycle. In general in the discussion section I found the distinction between malting and fermentation inconsistent – would be worth careful review and rewording.

Minor suggestions

43. The first two paragraphs of the abstract are essentially introductory text from the main text. I would suggest rewriting the abstract following the standard structure set out at: https://www.wiley.com/network/researchers/preparing-your-article/how-to-write-a-scientific-abstract

61. keywords – no need to repeat words already in the title.

107. Suggest reword “no scientifically clear answer to the question how the ancient beer was made” as it is not really a single question. Maybe “Our knowledge of the occurrence and manufacture of ancient beer is highly incomplete”?

117. Briefly define saccharification at this point.

136. The manuscript would benefit from copy-editing by a native English speaker, e.g. for text such as “The diagram bases on historical and ethnographic records” – better as “…is based on…”

153. For accessibility I suggest avoiding Latin terms (except when essential terminology)

167. ACO is only used twice in the text so I suggest it is discarded as an unnecessary abbreviation (from the keywords too).

403. Table 1 needs full caption.

6. PLOS authors have the option to publish the peer review history of their article (what does this mean?). If published, this will include your full peer review and any attached files.

Reviewer #1: No

Reviewer #2: Yes: Mark Nesbitt

---

## [Author Response · Author response to Decision Letter 0]

11 Mar 2020

Journal Requirements:

1. When submitting your revision, we need you to address these additional requirements. Please ensure that your manuscript meets PLOS ONE's style requirements, including those for file naming. The PLOS ONE style templates can be found at

Answer: We are aware of the style requirements and we have applied all style requirements in the revised version. In addition to the improvements of the manuscript according to the reviews, a few additional modi-fications have been carried out:

- Tables 1–4 have been moved to the Supplementary Information, as they are not required for the immediate understanding of the manuscript’s contents.

- The boxplot diagram of experimental malt (former Fig 14) has been removed as its entire content was also present in the overview boxplot diagram (current Fig 16). The latter now also shows the taxa identified in the respective objects.

- Affiliations have been updated, current addresses according to the PLOS Submission Guide-lines are now also included.

- Additional information concerning the variability of the chemical constituents of the aleu-rone cell wall has been included.

- Fig 18 has been added in order to illustrate the possible genesis of the Hornstaad find.

2. In your manuscript, please provide additional information regarding the specimens used in your study. Ensure that you have reported specimen numbers and complete repository information, in-cluding museum name and geographic location.

Answer: All necessary information is now represented in the manuscript.

- If permits were required, please ensure that you have provided details for all permits that were ob-tained, including the full name of the issuing authority, and add the following statement:

'All necessary permits were obtained for the described study, which complied with all relevant regu-lations.'

Answer: No permits were required, we have added the respective declaration.

For more information on PLOS ONE's requirements for paleontology and archaeology research, see https://journals.plos.org/plosone/s/submission-guidelines#loc-paleontology-and-archaeology-research.

"The authors have declared that no competing interests exist. ". We note that one or more of the au-thors are employed by a commercial company: 'BIAX Consult,' and 'Braxar GmbH'.

1. Please provide an amended Funding Statement declaring this commercial affiliation, as well as a statement regarding the Role of Funders in your study. If the funding organization did not play a role in the study design, data collection and analysis, decision to publish, or preparation of the man-uscript and only provided financial support in the form of authors' salaries and/or research materi-als, please review your statements relating to the author contributions, and ensure you have specifi-cally and accurately indicated the role(s) that these authors had in your study. You can update au-thor roles in the Author Contributions section of the online submission form.

Answer: Commercial affiliations have now been declared and justified in detail (see also in the Cover Letter).

“The funder provided support in the form of salaries for authors [insert relevant initials], but did not have any additional role in the study design, data collection and analysis, decision to publish, or preparation of the manuscript. The specific roles of these authors are articulated in the ‘author con-tributions’ section.”

Answer: The statement has been added as requested (see also in the Cover Letter).

Answer: The commercial affiliations of two of the authors did not play any role in our study, the respective statement has been added (see also in the Cover Letter).

2. Please also provide an updated Competing Interests Statement declaring this commercial affilia-tion along with any other relevant declarations relating to employment, consultancy, patents, prod-ucts in development, or marketed products, etc. 

Within your Competing Interests Statement, please confirm that this commercial affiliation does not alter your adherence to all PLOS ONE policies on sharing data and materials by including the fol-lowing statement: "This does not alter our adherence to PLOS ONE policies on sharing data and ma-terials.” (as detailed online in our guide for authors http://journals.plos.org/plosone/s/competing-interests) . If this adherence statement is not accurate and there are restrictions on sharing of data and/or materials, please state these. Please note that we cannot proceed with consideration of your article until this information has been declared.

Answer: The requested statements have been added (see also in the Cover Letter).

Please know it is PLOS ONE policy for corresponding authors to declare, on behalf of all authors, all potential competing interests for the purposes of transparency. PLOS defines a competing interest as anything that interferes with, or could reasonably be perceived as interfering with, the full and ob-jective presentation, peer review, editorial decision-making, or publication of research or non-research articles submitted to one of the journals. Competing interests can be financial or non-financial, professional, or personal. Competing interests can arise in relationship to an organization or another person. Please follow this link to our website for more details on competing interests: http://journals.plos.org/plosone/s/competing-interests

Reviewers' comments:

Reviewer's Responses to Questions

Comments to the Author

1. Is the manuscript technically sound, and do the data support the conclusions?

The manuscript must describe a technically sound piece of scientific research with data that sup-ports the conclusions. Experiments must have been conducted rigorously, with appropriate controls, replication, and sample sizes. The conclusions must be drawn appropriately based on the data pre-sented. 

Reviewer #1: Partly

Reviewer #2: Partly

 2. Has the statistical analysis been performed appropriately and rigorously? 

Reviewer #1: Yes

Reviewer #2: I Don't Know

3. Have the authors made all data underlying the findings in their manuscript fully available?

The PLOS Data policy requires authors to make all data underlying the findings described in their manuscript fully available without restriction, with rare exception (please refer to the Data Availa-bility Statement in the manuscript PDF file). The data should be provided as part of the manuscript or its supporting information, or deposited to a public repository. For example, in addition to sum-mary statistics, the data points behind means, medians and variance measures should be available. If there are restrictions on publicly sharing data—e.g. participant privacy or use of data from a third party—those must be specified.

Reviewer #1: Yes

Reviewer #2: Yes

4. Is the manuscript presented in an intelligible fashion and written in standard English?

Reviewer #1: Yes

Reviewer #2: No

5. Review Comments to the Author

Reviewer #1: This is an interesting manuscript that proposes the hypothesis that thinned aleurone layers of cereal grains are a marker for germination of cereals by design (malting) for making beer. The subject is certainly of interest in the cultural evolution of humans.

Data to support this hypothesis have been assembled from charred plant remains obtained from five archaeological sites in Egypt and Europe, and from model experiments. Contextual evidence of beer making was presented for the two sites in Egypt. The data collection and analysis seem thorough, the manuscript is clearly presented and well written, and it tells a cohesive story. Measurements of the width of aleurone cell layers in charred, sprouted barley grains (implicitly obtained under malt-ing conditions) in the model experiments demonstrated a decrease in thickness over time, consistent with processes that occur during malting. The contextual evidence from the sites in Egypt, and the importance of beer in the pre-historic culture of this region, adds some weight to the proposal that the Egyptian samples provide evidence for malting and beer making. However in my opinion, the evidence from the archaeological samples from Europe is too sketchy to be indicative of beer brew-ing, and the evidence as a whole is insufficient to support a general claim of finding a marker that “can be considered in the research for ancient beer in any archaeological site where charred preser-vation occurs”.

Most “ancient beer” markers are quite remote from the actual fermentation process within the chaînes opératoires of beer making (such as: archaeological finds of implements, finds of sprouted grains), and some are even arbitrary (as calcium oxalate precipitates). Yet the combination of differ-ent independent markers/indicators adds up to some rather sound evidence of the general practice.

Answer: We have followed the reviewer's recommendation and have removed all statements that might have given the impression of an equation of aleurone thinning to beer. What we wish to emphasize is that this feature represents an entirely new possibility of unlocking sprouting (which is a core process in beer making). It is now more clearly stated that “thin aleurone = sprouted grain”. This new marker significantly adds up to the extant toolkit of ancient beer research as it is entirely inde-pendent from any other evidence typically used for finding hints on beer production. We have also modified the text in order to allow alternative interpretations to thin aleurone cell walls, induced by other factors, e.g. sourdough production.

We did, however, broaden the perspective on malt-based foodstuffs in general in the section “’Egyp-tian ale’ and ‘lakeshore lager’? Take them with a pinch of… malt!”. Interpretations of the central European malt finds have been “softened” towards a more general nature. The need for extensive ethnographical and historical overviews of malt foods is now expressed.

Sprouted grains (most likely affected by microbial activity) have been used as foods in many cul-tures for as long as grains have been cultivated.

We have modified our text for clarity, explaining the mechanisms behind the cell wall degradation. We have laid this out clearly in multiple instances of the manuscript, thoroughly supported by sound arguments and solid references:

• section “Cell wall degradation as a consequence of sprouting”

• section “A diagnostic feature hidden in plain sight?”

• section “Possible pitfalls in the interpretation of thin-walled aleurone tissue”.

A statement that malting was associated with food processing might seem more reasonable than linking it strongly to beer production, although that may have been an outcome.

Answer: We explicitly state the general nature of our finds in the manuscript in multiple instances, such as:

• section “A new look at charred finds of cereal products”, line 179 f: “at least as an indirect additional marker” 

• section “A diagnostic feature hidden in plain sight”, line 213 f: “can serve as an indirect in-dicator of beer making “, and lines 229-230 f: “marker for malting, and albeit only indirectly pointing towards beer (…)”

• lines 626 f: “In contrast, the same aleurone thinning observed in the finds from the three central European Neolithic lakeshore settlements is not related to any clear evidence – be it contextual or artefactual – for brewing activities. So, how much of a hard proof for brewing do we really have?”

• and the following paragraphs

We have also broadened the perspective on malt-based foodstuffs in general in the section “’Egyp-tian ale’ and ‘lakeshore lager’? Take them with a pinch of… malt!”. The possible use of sourdough was also mentioned on several occasions in the text.

A greater concern I have relates to ancient cereal grains being much smaller than those from mod-ern varieties (e.g., Ferrio et al. 2006. J Cer Sci44, 323-332; Helbelstrup 2017. Plant Sci 256, 1-4).

Answer: We cannot agree with this claim and give following arguments:

1) We have added a discussion on aleurone composition and its relation to environmental con-ditions as well as to differences in species and landraces. However, the thickness of aleurone cell walls is apparently neither a function of grain size, nor of any of the aforementioned factors. The paper by Hands et al. 2012 e. g. demonstrates quite impressively that even small-seeded wild Poaceae do indeed have massive aleurone cell walls. Although we cannot completely exclude every other potential factor causing aleurone thinning (which is, of course, epistemologically impossible), our criterion has a strong diagnostic value within the given examples and is based on sound arguments and congruent observations.

2) In support of our arguments, we may bring to Reviewer`s 1 notice that the paper by Hebel-strup 2017 actually refers to wild versus domesticated forms, and to the beginnings of do-mestication in the Near East, at least 5,000 years earlier than in the sites mentioned in the current manuscript. We need not go into detail much more, but happily support the referee with a standard literature reference for further lecture (Zohary, D., M. Hopf, and E. Weiss, eds., 2012, Domestication of Plants in the Old World).

3) In the same supportive spirit, we may point out that the paper by Ferrio et al. 2006 uses grain weight on charred archaeological seeds, interpreting different grain sizes as conse-quences of growth conditions (!) and neither implies any (pre-)historical taxonomic traits nor a connection between grain anatomy/histology and grain size as suggested by the refer-ee.

The measurements taken from all of the archaeological samples, showed aleurone layers of similar thickness, but significantly smaller than those from the modern grains. This observation could be explained as structures coming from small grains rather than being indicative of germination.

Small grains even of other Poaceae species do not have thinner cell walls in general than e. g. culti-vated barley (see above). Triticum aestivum is among the exceptions, which has been dealt with in the manuscript. We have brought forward (see section “Cell wall degradation as a consequence of sprouting” and references cited there), that the progressive thinning of aleurone is a function of a) sprouting time, and b) the position of the observed aleurone within the germinating grain. We have explained these facts in the manuscript. For Reviewer 2, an outstanding researcher and number one expert in the field, all this was perfectly legible, comprehensible, and stringent in its argumentation.

However, following up Reviewer 1’s comments and Reviewer 2’s explicit request, we also added measurements of synchronous “reference” remains of ancient processed cereal foodstuffs displaying normal cell wall thicknesses in the aleurone layer. These samples may serve as a rebuttal to the aforementioned claim, and also as a data source strengthening the differentiation between different food types from the same site.

Grains are naturally associated with many active microbes, which as pointed out, could have caused the morphological changes. Hence, I am not persuaded by the statement of “thinned (degraded) aleurone cell walls as a general new marker for recognizing malting in cereal products from archae-ological contexts”.

Answer: We have elaborated in very detailed ways why microbial changes can be ruled out as the causes of cell wall thinning (see above).

We want to express our concern about Reviewer 1’s statement of being “not persuaded” without bringing forward any further substantial evidence or reason, aside from the claims which we have already refuted above. In our opinion, this is not sufficient for a qualified scientific debate.

Overall, I find the manuscript could make a useful contribution to this interesting area of prehistoric human activity if the discussion, conclusions and abstract are recast and softened to be more bal-anced to alternative interpretations. The title is cumbersome and should be shortened … at the least, I suggest omitting the words “and comprehensive” and the words after “record”.

Answer: We have extensively elaborated on all possible and probable alternative interpretations (also ac-cording to Reviewer 2). We have thoroughly explained why we think that some of them can work, and others cannot.

We have followed Reviewer's 1 recommendations and thus rephrased several statements in the ab-stract, discussion and conclusions in order to achieve more balanced appearance of our manuscript, and “soften” some of the argumentation.

The title has now been shortened.

I also recommend the following minor editorial corrections.

1. The statement that the main protein content in cereal grains is in the aleurone layer (p6, line 185) is incorrect; aleurone cells contain protein bodies, but the majority of cereal grain proteins are stor-age proteins in the endosperm.

Answer: We have modified this accordingly.

2. The term “heterocelluloses” is inaccurate and not meaningful; it should be replaced by heteropol-ysaccharides.

Answer: We have exchanged “heterocelluloses” by the term “hemicelluloses” which is even more accurate and meaningful than “heteropolysaccharides”.

Reviewer #2: This paper presents an important advance in the detection of malting in ancient sam-ples, moving beyond starch characteristics that usually disappear during charring. The aleurone layer often survives charring and is recognisable under SEM, meaning that the proposed criterion can be widely applied to charred and uncharred residues. The ability to detect malting in a wide range of archaeological contexts has the potential, as demonstrated through the case studies, to sig-nificantly advance our understanding of the dietary and social roles of food in ancient societies.

The paper is well-grounded in the literature of brewing science and extensive experimentation. Data is fully presented in the supplementary files and technical aspects of methodology are fully ex-plained. Alternative explanations are fully considered. Although I make some suggestions on meth-odology below, I would expect the requested additional information to confirm the results of this study.

My suggestions for major changes are as follows:

Control samples:

- For the experimental work, I could not see the figures for aleurone thickness for the Heinrich Durst Malzfabriken barley grains *before* malting, either in table 1 or elsewhere – did I miss them? In the case of wheat, where there are control (unmalted) grains and malted grains (line 274), I could not find the aleurone thickness reported in the paper. For both wheat and barley it’s hard to assess the evidence for the effects of malting without seeing the measurements for malted vs unmalted grains.

Answer: The basic aim of the paper is not to demonstrate that malting has any effect on cereal aleurone cell wall thickness, as the basic research on this topic has been carried out decades ago (cf. introductory chapters). Our goal was to raise awareness, and significance, of this phenomenon in archaeobotani-cal materials, and to test whether the cell wall thinning can also be observed in charred state. This has been explained more clearly now in the research goals section.

We left out entirely unmalted barley grains due to the fact that measurable loss in cell wall sub-stance is not observable in the first day of malting. This has now been more explicitly stated in the section “Experimentally charred malt”.

- For the ancient residues, it should be standard practice to compare residues from ‘likely’ contexts with those from ‘unlikely’ contexts. For example, when evidence of food only occurs in association with food-related objects and is absent from tested non-food related objects or soil matrix, it strengthens the case for the age and identification of the food residue. In this instance the control sample could be grains from the sites or periods concerned that are intact and do not show signs of germination. Would it be possible to present some data on aleurone thickness from intact charred grains from the European sites? And for the Egyptian sites (material harder to access) at least from uncharred grains of a similar date in accessible collections. As with the experimental samples, com-paring measurements for likely-malted vs unlikely-malted would strengthen the authors’ case.

Answer: For two of the three central European sites (Sipplingen and Zürichsee), other processed food resi-dues are available which do not show the cell wall thinning, and therefore likely derive from un-malted grains. These are now included into the data, their characteristics are explained.

Beer vs malted foods

135, Fig. 1 and various points in the manuscript. I think you could strengthen the distinction be-tween mashing and fermenting (probably should be distinct steps in Fig 1). *It is only the last step that produces alcohol and is evidence for beer.* The criterion of the thinning of the aleurone layer is, if I understand correctly, evidence of saccharification, not of fermentation?

Saccharification (during malting) is now more explicitly shown as a separate process in Fig. 1. How-ever, saccharification does continue during fermentation, which has now also been clarified in Fig 1. The aleurone thinning is a marker for malting/sprouting/saccharification. This is now more clearly explained in the text in section “Defining core processes of beer making”.

 - It might be worth adding a heading to explore the circumstances in which evidence of malting and grinding can be interpreted as evidence of beer. Of course human nature and Occam’s razor sug-gests this is often the case, but at the same time there is a risk inherent in projecting our current in-terest in beer back into the past. For example, in the paragraph starting 503, for example, one residue has a liquid appearance, leading to all being interpreted as beer. But couldn’t this be a malt drink? And the non-liquid food, malt foods? There is a strong tradition in several parts of the world of malted foods or beverages, notably in connection with weaning, and with millets in Africa. It would be worth adding a paragraph to explore this – there is plenty of literature on malted foods. What does the record look like for traditional food cultures in Europe? If malted foods and drinks are ab-sent it strengthens the case for beer. Adam Maurizio might be good for this.

Answer: We have added examples of other malt-based food than beer from the (ethno-)historical literature, stating that malt finds lacking a clear brewing-associated context will require more caution for their interpretation. Conclusions towards beer have been rephrase now for the central European malt finds. We now are pointing out that a concise overview of malted foodstuffs in the Old World is a big desideratum for research, in order to provide a reliable source of comparative data for archaeo-logical remains of malt-based foodstuffs.

Another example is at 523 “As beer brewing based on sprouted grains is ubiquitous in the ethno-graphic, historical, and archaeological record, probability is very high that these remains indeed represent Neolithic beer mash.” is another example of projecting the ubiquity of beer today into the archaeological record, creating a self-fulfilling cycle. In general in the discussion section I found the distinction between malting and fermentation inconsistent – would be worth careful review and rewording.

Answer: We have changed this accordingly (see similar issue raised above).

Minor suggestions

43. The first two paragraphs of the abstract are essentially introductory text from the main text. I would suggest rewriting the abstract following the standard structure set out at: https://www.wiley.com/network/researchers/preparing-your-article/how-to-write-a-scientific-abstract

Answer: The abstract has been rewritten accordingly.

61. keywords – no need to repeat words already in the title.

Answer: The title has been shortened, now there are no more overlaps with the keywords

107. Suggest reword “no scientifically clear answer to the question how the ancient beer was made” as it is not really a single question. Maybe “Our knowledge of the occurrence and manufacture of ancient beer is highly incomplete”?

Answer: Changed accordingly.

117. Briefly define saccharification at this point.

Answer: Changed accordingly.

136. The manuscript would benefit from copy-editing by a native English speaker, e.g. for text such as “The diagram bases on historical and ethnographic records” – better as “…is based on…”

Answer: Language editing by a native English speaker has been carried out.

153. For accessibility I suggest avoiding Latin terms (except when essential terminology)

Answer: See above.

167. ACO is only used twice in the text so I suggest it is discarded as an unnecessary abbreviation (from the keywords too).

Answer: We have placed the acronym into the paper on purpose: There will be a paper on diagnostic ap-proaches towards amorphous charred objects (ACO), and we would really like to promote this acro-nym as parts of a coming “standard terminology” for such kinds of material. Following the review-er’s suggestions, we have now included ACO in multiple instances.

403. Table 1 needs full caption.

Answer: The tables have been partly merged, partly moved into the Supplementary materials. We hope all captions are now satisfactorily written.

6. PLOS authors have the option to publish the peer review history of their article (what does this mean?). If published, this will include your full peer review and any attached files.

Do you want your identity to be public for this peer review? For information about this choice, including consent withdrawal, please see our Privacy Policy.

Reviewer #1: No

Reviewer #2: Yes: Mark Nesbitt

While revising your submission, please upload your figure files to the Preflight Analysis and Con-version Engine (PACE) digital diagnostic tool, https://pacev2.apexcovantage.com/. PACE helps en-sure that figures meet PLOS requirements. To use PACE, you must first register as a user. Registra-tion is free. Then, login and navigate to the UPLOAD tab, where you will find detailed instructions on how to use the tool. If you encounter any issues or have any questions when using PACE, please email us at figures@plos.org. Please note that Supporting Information files do not need this step.

---

## [Editor Report · Decision Letter 1]

31 Mar 2020

Mashes to Mashes, Crust to Crust. Presenting a novel microstructural marker for malting in the archaeological record

PONE-D-19-25202R1

Dear Dr. Marinova,

We are pleased to inform you that your manuscript has been judged scientifically suitable for publication and will be formally accepted for publication once it complies with all outstanding technical requirements.

With kind regards,

Ceren Kabukcu, PhD

Academic Editor

PLOS ONE

Additional Editor Comments (optional):

Dear Authors,

I hope you are all safe and healthy under the exceptional circumstances we find ourselves. I have reviewed the article and your replies to reviewer comments and recommend that the article can go ahead with publication.

All best wishes,

Ceren
---

## [Editor Report · Acceptance letter]

10 Apr 2020

PONE-D-19-25202R1 

Mashes to Mashes, Crust to Crust. Presenting a novel microstructural marker for malting in the archaeological record 

Dear Dr. Marinova:

I am pleased to inform you that your manuscript has been deemed suitable for publication in PLOS ONE. Congratulations! Your manuscript is now with our production department. 

With kind regards,

on behalf of

Dr. Ceren Kabukcu 

Academic Editor

PLOS ONE